# Probabilistic characterisation of coastal storm-induced risks using Bayesian Networks

Marc Sanuy[1], Jose A. Jiménez[1]

[1]Laboratori d'Enginyeria Marítima, Universitat Politècnica de Catalunya, BarcelonaTech, c/Jordi Girona 1-3, Campus Nord ed D1, 08034 Barcelona, Spain.

*Correspondence to*: Marc Sanuy (marc.sanuy@upc.edu)

**Abstract.** A probabilistic estimation of hazards based on the response approach requires assessing large amounts of source characteristics, representing an entire storm climate. In addition, the coast is a dynamic environment, and factors such as existing background erosion trends require performing risk analyses under different scenarios. This work applies Bayesian Networks (BNs) following the source-pathway-receptor-consequences scheme aiming to perform a probabilistic risk characterisation at the Tordera Delta (NE Spain). One of the main differences of the developed BN framework is that it includes the entire storm climate (all recorded storm events, 179 in the study case) to retrieve the integrated and conditioned risk-oriented results at individually identified receptors (about 4,000 in the study case). Obtained results highlight the storm characteristics with higher probabilities to induce given risk levels for inundation and erosion, and how these are expected to change under given scenarios of shoreline retreat due to background erosion. As an example, storms with smaller waves and from secondary incoming direction will increase erosion and inundation risks at the study area. The BNs also output probabilistic distributions of the different risk levels conditioned to given distances to the beach inner limit, allowing for the definition of probabilistic setbacks. Under current conditions, high and moderate inundation risks, and direct exposure to erosion can be reduced with a small coastal setback (~10 m), which needs to be increased up to 20-55 m to be efficient under future scenarios (+20 years).

## 1. Introduction

The coastal fringe is a highly dynamic zone and one of the most fragile terrestrial areas due to high population, dense infrastructure, intense economic activities, and endangered natural habitats. The progressive occupation of coastal areas increasingly exposes them to storm-induced hazards, such as inundation and erosion (IPCC, 2012, 2013). This, together with future projections of rising sea levels (Vousdoukas et al., 2016; IPCC, 2018), long-term shoreline retreat (Vousdoukas et al. 2020), changes in storminess (Lionello et al. 2008, Conte and Lionello 2013; IPCC, 2014), and/or changes in the directionality of incoming waves (Cases-Prat and Sierra, 2013), highlight the need for local-scale risk assessments considering these current and future scenarios. In the NW Mediterranean basin, storm-induced damages at the Catalan coast have increased during the last decades as a result of increased exposure along the coastal zone and the progressive narrowing of the existing beaches (Jiménez et al., 2012). All these elements have determined that current and future coastal management plans will require a specific chapter on coastal risks as recognised in the Protocol of Integrated Coastal Zone Management in the Mediterranean (UNEP/MAP/PAP, 2008). One of the most used approaches in risk assessment is the Source-Pathway-Receptor-Consequences (SPRC) framework (Sayers et al., 2002; Narayan et al. 2014; Oumeraci et al., 2015). This is a conceptual model describing the propagation of risk across a given domain from the source to the receptors. When applied to storm-induced coastal risks, it is generally schematised in terms of a source (storms), that propagates and interacts with a pathway (beach or coastal morphology) where hazards (i.e. inundation and erosion) are generated. These affect the receptors (elements of interest), inducing different consequences. When addressing the problem at the local scale (~5–10 km), storm-induced hazards are usually assessed by using detailed process-based models that are fed information on

both the source and the pathway. Recent studies use the capabilities of Bayesian Networks (BNs) to assess consequences at the receptor scale, as they can easily handle multidimensional problems while dealing with large amounts of data allowing the assessment of multiple source conditions, hazards, and scenarios (e.g. Van Verseveld et al., 2015; Poelhekke et al., 2016; Plomaritis et al., 2018; Sanuy et al., 2018). BNs allow the analysis of conditional dependencies between variables, and therefore, can be used to reproduce the causal relationships inherent in the SPRC scheme (Jäger et al., 2018).

In this context, this work presents the development of a fully probabilistic BN-based SPRC approach to assess storm-induced risks at a local scale. To illustrate the methodology, the BN approach is applied to characterise coastal risks at the Tordera Delta, a highly dynamic area that is vulnerable to the impact of extreme coastal storms (Jiménez et al., 2018). Risks related to storm-induced erosion and inundation were assessed using current morphology and future configurations considering the existing trends of shoreline retreat due to background erosion (Jiménez et al., 2019). The approach assesses the storm characteristics associated with the spatially variable risks, and characterises the along-shore and cross-shore spatial distribution of given levels of risk under different scenarios. For this purpose, all available storms derived from a long dataset (60 years) of wave time series were simulated by the XBeach model (Roelvink et al., 2009) and the induced hazards analysed. Receptor characterisation was individually performed as described in Sanuy et al. (2018). The inundation risk was assessed in terms of relative damage to structures and risk to life, while the erosion risk was assessed as a function of the loss of protective capacity of the coast in front of the receptors. The inclusion in the BN of simulation results from a long dataset of storms allows for a fully stochastic assessment in terms of wave climate characterisation. This is a novelty with respect to existing studies (e.g. Van Verseveld et al., 2015; Plomaritis et al., 2018; Ferreira et al. 2019; Sanuy et al., 2018). Although some of these studies introduce copula assessments on source (storm) characteristic variables to generate synthetic events, the training subsets aimed to covering the whole range of possible storm conditions rather than statistically representing the existing storm climate. In addition, the applied method follows the idea behind the response approach (Garrity et al., 2006, Sanuy et al., 2020a), simulating erosion and inundation hazard for the whole population of events, while simulating the storms using their real shapes (i.e. storm evolution with time), and thus, avoiding the uncertainties introduced by the use of a synthetic representation of the events (Duo et al., 2020).

The structure of the paper is as follows: Section 2 presents the study area with the main data sources, Section 3 outlines the methodology and its different steps, and Section 4 presents the obtained risk characterisation at the Tordera Delta; results are discussed in Section 5 and the main conclusions are summarised in Section 6.

## 2. Study area and data

The Catalan coast is located in the NW Mediterranean Sea (Figure 1). The coastline extends to nearly 600 km with about 280 km of beaches. Storm-induced issues are present along the entire coastline and are especially concentrated in locations with the largest decadal-scale shoreline erosion rates (Jiménez et al., 2011; Jiménez and Valdemoro, 2019). A good example of such an area is the Tordera Delta, located approximately 50 km north of Barcelona (Jiménez et al., 2018) (Figure 1). The deltaic coast is a highly dynamic area composed of coarse sediment and extends to about 5 km, from s'Abanell beach at the northern end to Malgrat de Mar beach in the south (Figure 1). It is currently retreating because of the net longshore sediment transport directed southwest and the decrease in Tordera River sediment supplies. Consequently, the beaches surrounding the river mouth are being significantly eroded (Jiménez et al., 2011; Sardá et al., 2013; Jiménez et al., 2018), and the frequency of inundation episodes and damage to existing infrastructure (beach promenade, campsite installations, roads, etc.) has significantly increased since the beginning of the 90s (Jiménez et al., 2011; Sardá et al., 2013) (Figure 1). The area is composed of multiple campsites that represent the main economical activity of the municipality and was identified as a regional coastal hotspot to erosion and inundation in Jiménez et al., (2018). Therefore, it is the prototype of study area where detailed risk assessments are needed at the local scale to support decision making.

To spatially characterise the risk of the area as a function of the variability of the local geomorphology and coastline orientation at both sides of the river mouth, five different sectors along the coast were defined (Figure 1). Two of them, SBN and SBM, are located northwards of the river mouth (Figure 1), with SBM being limited to the south by the river mouth. The main distinctive feature of SBN is the existence of a promenade limiting the inner part of the beach. Southwards of the river, there are three sectors (Figure 1): MSM being closest to the mouth; MS1, which is located southwards of a coastal

revetment; and MS2 located furthest to the south, with wider beaches, and sheltered against Eastern storm waves, which are dominant in the area (Mendoza et al. 2011).

The data used to represent the morphology of the study area are comprised of LIDAR-derived topography provided by the Institut Cartogràfic i Geológic de Catalunya, as a high-resolution digital elevation model (DEM) with 1-m × 1-m grid cells and a vertical precision of 5–6 cm (Ruiz et al. 2009). Bathymetry obtained from multi-beam surveys provided by the

Ministry of Agriculture, Fish, Food, and Environment was also used.

To characterise the forcing, the present work used hindcast waves from the Downscaled Ocean Waves dataset (Camus et al., 2013) derived from the Global Ocean Waves (Reguero et al., 2012). Hindcast surge from the Global Ocean Surge dataset (Cid et al. 2014), obtained at 4 locations close to the Tordera Delta at ~20 m depth, covering the period from 1954–2014 (Figure 1), was also used. The simultaneous astronomical tide was added to the Global Ocean Sampling (GOS) dataset to

obtain the total water level. The astronomical tidal range in the study area was about 0.25 m.

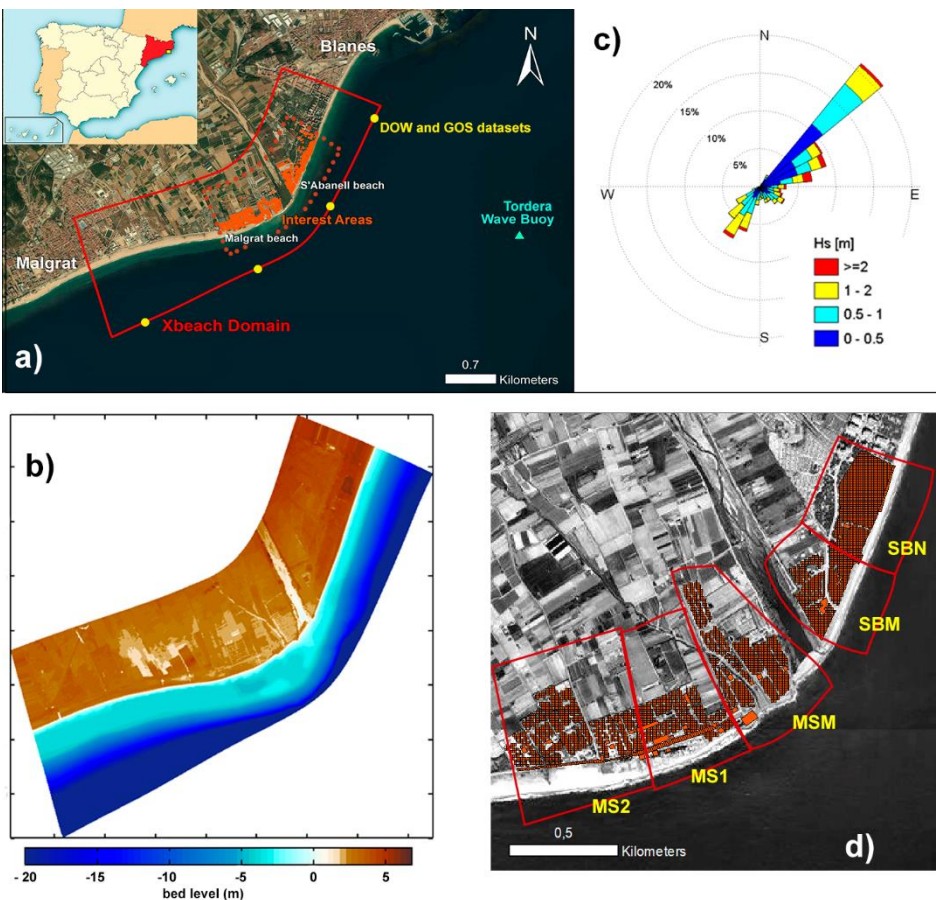

Figure 1: Main locations and characteristics of the study site: a) Location of the Tordera Delta, XBeach model domain (red), location of model boundary conditions (yellow, Downscaled Ocean Waves, and Global Ocean Surge datasets, Camus et al., 2013), receptors of interest (orange) and Tordera wave buoy (light blue); b) Digital Elevation Model (DEM) of the Tordera Delta; c) wave rose at the Tordera delta buoy (Global Ocean Waves; Reguero et al., 2012); d) receptor areas for the local risk assessment. Orthophoto provided by Institud Cartogràgic I Geològic de Catalunay (ICGC).

## 3 Methodology

### 3.1 General framework

The methodology used in this work adapts the general approach of Jäger et al. (2018) where BNs were applied to implement the SPRC framework to assess storm-induced coastal risks. This approach has been previously implemented by Sanuy et al. (2018) at the Tordera Delta to compare, in a deterministic manner, different risk reduction measures. In this work, the scheme was upgraded to a fully probabilistic risk characterisation and consisted of the following steps:

(i) *Storm characterisation*. This step consisted of defining the local storm climate from long-term wave time-series. This

stage corresponded to the (probabilistic) characterisation of the source. In practice, the result of this step was a storm dataset containing the hourly evolution of wave parameters during each event for a long period (multiple decades).

(ii) *Hazard assessment*. Once the forcing was characterised, the next step was the assessment of the storm-induced hazards, i.e. erosion and inundation, which were simulated using a process-based morphodynamic model, XBeach. This stage corresponded to the characterisation of the pathway. To ensure the probabilistic representation of the

hazards, this step was performed for all the events of the storm dataset (first step) or for a subset of events that ensures an equivalent representation of the multivariant population representing the source.

(iii) *Risk characterisation*. In this step, simulated storm-induced hazards across the study area were transformed into risk values at the scale of individual receptors (existing buildings and infrastructure). To this end, vulnerability rules were defined as a function of the receptor typology and analysed hazard. In this stage, the receptor and consequence phases

of the SPRC framework were tackled.

(iv) *Scenario definition*. This step consisted of defining the conditions for the assessment in terms of geomorphological scenarios of interest. This might require repeating steps (ii) and (iii), for all identified storms in (i). Here, the entire storm dataset was used to characterise the baseline scenario (current conditions), while the additional scenarios were assessed with a representative subset to reduce the computational effort. The subset was also used to assess the

baseline scenario to later verify that it statistically represents the same population as the original dataset from the perspective of the obtained results (for the validation of the method).

(v) BN *integration*. The obtained results for each event at the receptor scale were related to the variables characterising the storms (e.g. bulk features) and receptor properties (e.g. location) and integrated within the BN. Therefore, the BN outputs risk probability distributions accounted for the variability in the forcing conditions as well as the spatial

distribution of receptors.

In the following sections, specific methods used in each step to analyse the Tordera delta case study are presented. Although some specificities are included, adopted methods are general enough to be applicable at nearly any site.

### 3.2 Storm characterisation


Coastal storms have been identified from wave time-series by employing the peak-over-threshold (POT) method using a double threshold criterion as in Sanuy et al. (2020a). The first threshold, the 0.98 quantile (Hs = 2 m, in agreement with Class 1 storms in Mendoza et al. 2011 for NW Mediterranean conditions), is used to identify storm start and end times, and thus, controls the event duration and inter-event fair-weather periods. The second threshold, the 0.995 quantile (Hs = 2.6 m),

is used to filter events that do not reach this value at the peak and would not be significant in terms of induced impacts. This second threshold retains only storms reaching Class 3 at the peak, which is the minimum storm magnitude inducing hazardous coastal response (Mendoza et al., 2011)

The obtained dataset is composed of 179 storms (~3 storms per year), each being characterised by the hourly evolution of wave conditions (significant wave height, Hs; peak period, Tp; storm surge; wave direction; and directional spreading). Of the 179 events, 43 correspond to multi-peak storms. These events occur when fair-weather conditions (Hs below the first threshold) between consecutive peaks last less than 72 hours (Figure 2); they are relatively frequent in this part of the NW Mediterranean (Mendoza et al. 2011). In 12 cases, storms are formed by 3 or more peak sequences, leading to a total number of 237 individual storm peaks. For each peak, we retain its duration, together with the total accumulated event duration, and the previous energy (e.g. single-peak storms are always characterised as peaks with "peak duration" equal to "event duration" and with "zero previous energy"). Although all this information is retained (Figure 2), only event duration together with wave parameters and water level will be used as BN variable here, for the sake of simplicity in a risk-oriented perspective, while more detailed source description may be necessary in morphological analyses.

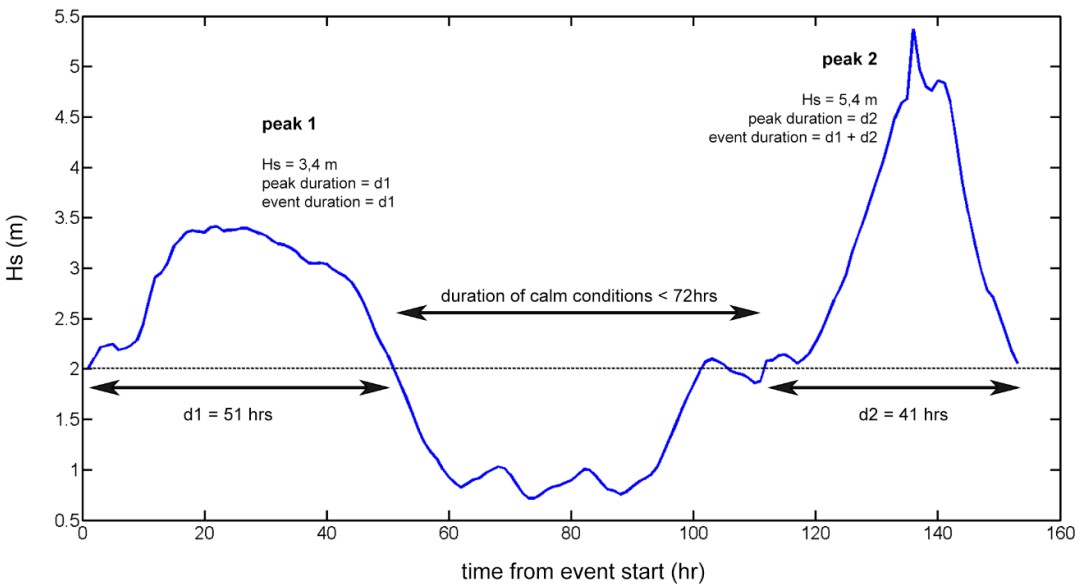

**Figure 2: Scheme of a double peak storm.**

To reduce the computational effort when assessing multiple scenarios, a storm *subset* is built aiming to maintain the statistical representativeness while avoiding the repetition of simulations of strongly similar storm conditions. The procedure consists of grouping the main variables defining the storm (Hs, Tp, duration, and direction) in homogeneous intervals covering the entire range of local conditions (see Table 1). Each storm from the dataset falls into one of the resulting $5 \times 4 \times 3 \times 3 = 188$ combinations of bulk characteristics. Some combinations are populated with storms (48), while others are empty groups (140), i.e. storm characteristics that have not been recorded and, therefore, not present in the storm dataset. This subdivision is only used for the purpose of deriving the subset, allowing finer detail in the source characteristics of the single-peak and multi-peak storms to be selected. Later, the BN will present a coarser binning of such variables, ensuring a better filling of the source variable combinations in the network.

Therefore, to produce the subset, one storm is selected for each combination populated with at least one event. To ensure a probabilistic representation of the source, the number of storms belonging to each combination is counted for later use as a weight (multiplicity factor) when feeding the BN with results from that event.

**Table 1: Subset characteristics compared to the original storm dataset. Source variable combinations used to classify storms and select the subset events.**

| Original dataset characteristics | | | |
| --- | --- | --- | --- |
| **179 storms** | 136 single-peak | 43 multi-peak | 237 storm peaks |
| Subset characteristics | | | |

| 69 storms | 26 single-peak | 43 multi-peak | 127 storm peaks |
|---|---|---|---|
| **Variable combinations to produce subsets** | | | |
| **Hs (m)** | Tp (s) | Duration (h) | Direction (ºN) |
| **< 3** | < 9 | < 20 | > 110 |
| **3 – 3.5** | 9 – 11 | 20 – 40 | 110 – 150 |
| **3.5 – 4** | > 11 | 40 – 60 | > 155 |
| **4 – 4.5** | | > 60 | |
| **> 4.5** | | | |


As was previously mentioned, one of the local characteristics of the storm climate in the study area is the presence of multi-peak storms. As the impact of successive storms separated by relatively short fair-weather periods may be different to that of single events depending on storm characteristics and initial beach configuration (e.g. Dissanayake et al. 2015; Eichentopf et al. 2020), we retained these storms in the analysis. Thus, to properly account for their potential effects, all existing identified

multi-peak storms in the original time-series (43) were included in the *subset*. Their impact was simulated with the XBeach model saving the cumulative output after each peak. The impact after the first peak of such multi-peak events was used as proxy of equivalent single-peaks already covering 22 source variable combinations. The other 26 combinations where covered by additional single-peak storms.. Thus, the storm *subset* comprised of 69 storms, including the 43 multi-peak storm events (see Table 1).

The statistical representativeness of the *subset* with respect to the full storm dataset was tested using the methodology to compare histograms proposed by Bityukov et al. (2013). This method assumes that values at each bin of the histogram follow a normal distribution with expected value $n_{i,k}$ and variance $\sigma^2_{i,k}$ (with *"i"* representing the bin and *"k"* the histogram). Thus, the significance is defined as:

$$\hat{S}_i = \frac{\hat{n}_{i,1} - \hat{n}_{i,2}}{\sqrt{\hat{\sigma}_{i,1} + \hat{\sigma}_{i,2}}}, \tag{1}$$

where $\hat{n}_{i,k}$ is an observed value at bin *"i"* of histogram "k" and $\hat{\sigma}_{i,k} = \hat{n}_{i,k}$. Therefore, we consider the root mean square (RMS) of the distribution of significances as:

$$RMS = \sqrt{\frac{\sum_{i=1}^{M} (\hat{S}_i - \bar{S})^2}{M}}, \tag{2}$$


where $\bar{S}$ is the mean value of $\hat{S}_i$ and M is the number of bins of the histogram. The RMS represents a distance measure with the following interpretation: If RMS = 0, both histograms are identical; if RMS =0~1 both histograms are obtained from the same parent population; if RMS >> 1, histograms are obtained from different parent distributions. The method is applied to compare the output distributions resulting from training the BN with the whole dataset vs. training it with the subset.

The statistics will be calculated for both BN inputs and outputs (see following sections): (i) the distribution of un-constrained output risk variables; (ii) the distribution of Hs, Tp, duration, direction and water level constrained to the different risk levels per sector; and (iii) the risk distributions per area and conditioned to the distance to inner beach limit. This involves the comparison of more than one variable output (e.g. impact results are always three variables), and therefore, results are given as a mean and standard deviation.

## 3.3 Hazards assessment

Storm-induced hazards (erosion and flooding) have been modelled using the XBeach model (Roelvink et al. 2009), which has been previously calibrated for the Tordera Delta (see Sanuy et al. 2019b). The calibration of the model achieved a Brier Skill Score (BSS) (Sutherland et al. 2014) of 0.68. The model was implemented using a curvilinear grid with a variable cell size around the Tordera River mouth (Figure 1). The extension of the mesh is approximately 1.5 km in the cross-shore direction, with a cell size ranging from 5–6 m at the offshore boundary (20 m depth) to 0.7–0.8 m at the swash zone. In the alongshore direction, the model has an extension of 4.5 km with cell size ranging from 25 m at the lateral boundaries 2–3 m around the river mouth. Storm input consists of time-series of wave conditions characterising each storm obtained from the DOW dataset at the 4 nodes at the offshore boundary (Figure 1), with a time-step of 1 hour, which is the time resolution of the original data. The model was used to simulate storm-induced hazards under 455 different events, which correspond to 179 original storms, plus a *subset* composed of 69 storms under 4 different scenarios. The XBeach model outputs used for the subsequent risk calculations were *maxzs* for water depth with accompanying *u,v* components of the water velocity (inundation hazard) and *sedero* for bed level change (erosion hazard).

## 3.4 Risks

To assess the induced risk, first, receptors in the study area are individually considered by their footprint polygons (~4000) and delineated using a Geographic Information Systems (GIS)-based tool to account for their exact position and dimension. Once they are defined, a direct correspondence between each receptor with the underlying XBeach model mesh is available in such a way that each receptor is associated with the model nodes directly affecting it (see Figures 3 and 4).

The vulnerability of each receptor is individually characterised as a function of their structural properties. Receptors in the study area comprise hard constructions, such as houses and infrastructures, and softer elements such as campsite elements (e.g. bungalows) (Sanuy et al., 2018). To assess the flooding-induced risk, the relative damage to receptors is calculated using flood-damage curves (Table 2) using the maximum-modelled water depth within the receptor polygon. No specific damage curves are available for the Catalan coast, and due to this, we used the curves recommended and used by the Catalan Water Agency (ACA, 2014) for the development of inundation management plans. Risk to life has also been included in the assessment by using the water-depth-velocity product as input (Table 3, Priest et al., 2007) within the receptor's boundaries. For the erosion hazard, the magnitude of the associated risk is based on the distance from the significantly eroded XBeach nodes to the receptors. Significant erosion was set to 0.25 m of the vertical bed level change and assumed as the common minimum depth for light structure foundations. The closest distance from the receptor corners to that erosion level was compared with the erosion risk thresholds indicated by Jiménez et al. (2018) (Table 4).

Therefore, the result of each simulation (hazard maps) was transformed into a risk value at the individual receptor. Figure 3 shows an example of simulated inundation water depth for a long return period event and its transformation into relative inundation damages to receptors: None (0%), Low (0–30%), Moderate (30–60%), High (60–90%), and Extreme (>90%). Figure 4 shows, for the same event, results corresponding to the erosion hazard. Individual results were stored at each of the ~4000 receptors for each of the simulated events, leading to a total number of 716,000 and 276,000 cases to feed the BN with the entire dataset and with the *subset,* respectively.

**Table 2: Flood damage curves to obtain relative damage to structures using simulated inundation depth as input (Catalan Water Agency, ACA, 2014).**

| INUNDATION DEPTH (M) | RELATIVE DAMAGE (%) | |
|---|---|---|
| | **Hard structures** (Road, promenade, houses) | **Soft structures** (Campsite elements) |

| 0 | 0 | 0 |
|---|---|---|
| 0 – 0.3 | 18.3 | 50 |
| 0.3 – 0.6 | 26.5 | 71 |
| 0.6 – 0.9 | 33.2 | 82 |
| 0.9 – 1.5 | 44.7 | 89 |
| 1.5 – 2.1 | 54.9 | 91 |
| > 2.1 | 64.5 | 100 |

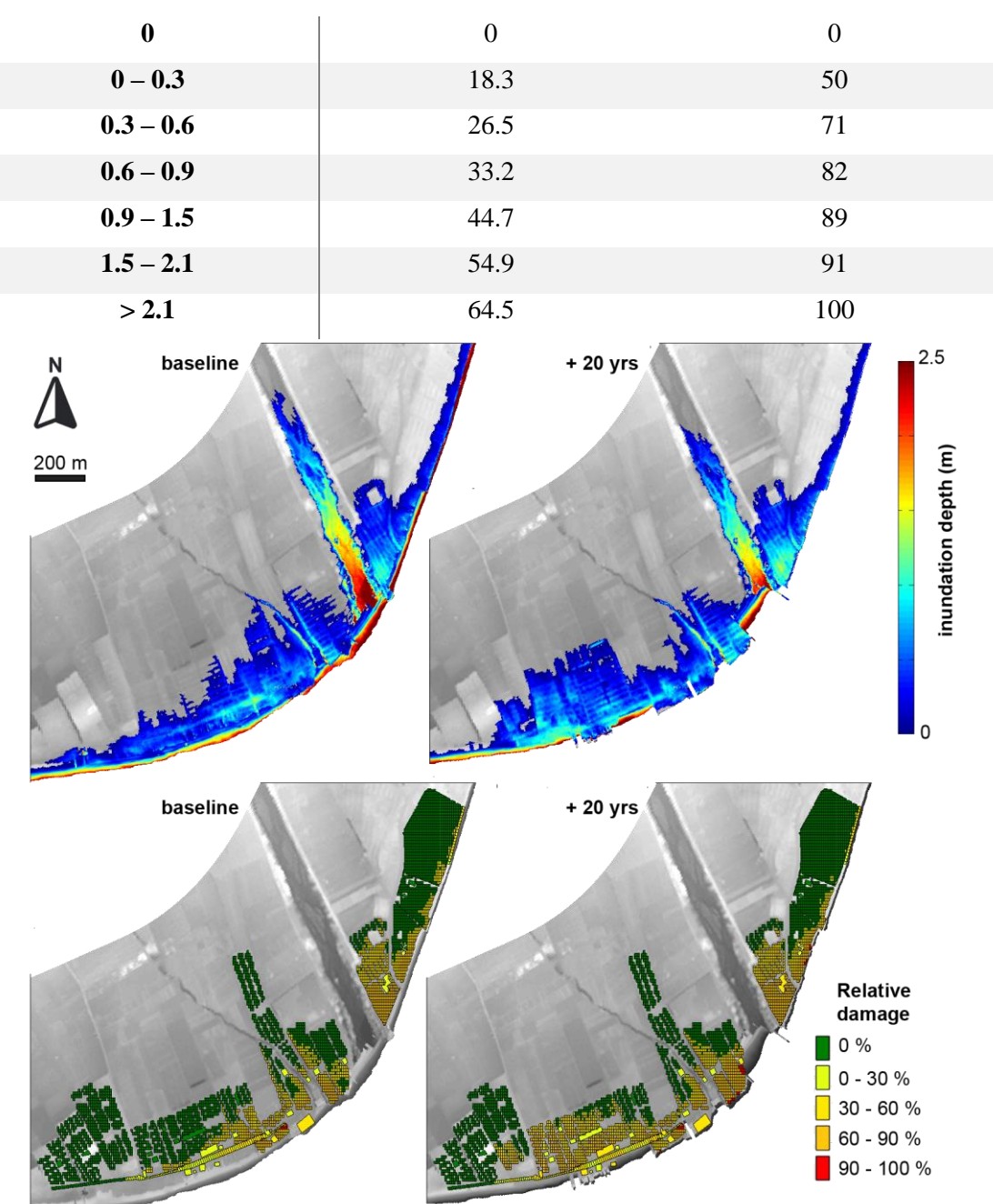

**Figure 3: Example of transformation from inundation hazard to risk. Storm event of November 2001, Hs = 5.4 m, Tp = 13 s, eastern direction, and 96 h of event duration. LIDAR provided by Institud Cartogràgic I Geològic de Catalunay (ICGC).**

**Table 3: Risk to life calculated as a function of the product between water depth and flow velocity (Priest et al. 2007).**

| Flood depth-velocity (m²/s) | Risk to Life |
|---|---|
| 0 – 0.25 | None |
| 0.25 – 0.5 | Low |
| 0.5 – 1.1 | Moderate |
| > 1.1 | High |

**Table 4: Erosion risk as a function of the distance from the receptors to erosion magnitudes greater than 0.25 m of bed level change. A distance of 7.5 m corresponds to the expected retreat for the 10-year return period (Jiménez et al., 2018).**

| Erosion risk level | Distance to receptor (m) |
|---|---|
| None | > 30 |

| Very Low | 22.5 – 30 |
|----------|-----------|
| **Low** | 15 – 22.5 |
| **Moderate** | 7.5 – 15 |
| **High** | 3 – 7.5 |
| **Extreme** | 0 – 3 |

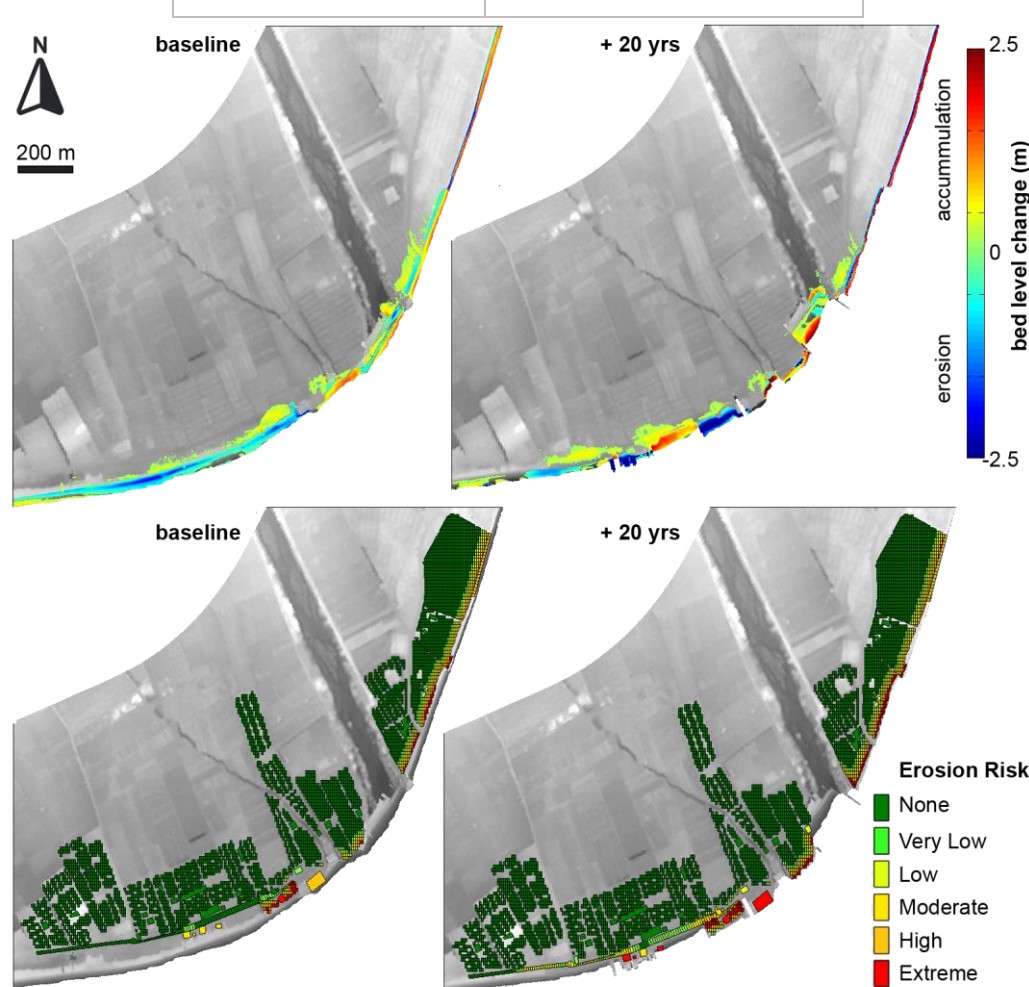


**Figure 4: Example of transformation from erosion hazard to risk. Storm event of November 2001, Hs = 5.4 m, Tp = 13 s, eastern direction, and 96 h of event duration. Orthophoto provided by Institud Cartogràgic I Geològic de Catalunay (ICGC).**

### 3.5 Scenario definition

When assessing risks in coastal areas under changing conditions, it is necessary to consider these potential variations in the assessment, otherwise, its utility for medium-long term risk management will be limited. Here, future morphological scenarios are defined to consider the background erosion in the area. As previously mentioned, the study area is a highly dynamic sedimentary environment subjected to a background coastal retreat (Jiménez et al. 2018). Thus, in this step, different scenarios characterising future configurations were built based on the expected future coastal changes. This was

accomplished by using decadal-scale background erosion rates estimated for the different beach sectors by Jiménez and Valdemoro (2019) by analysing shoreline changes from aerial photographs. The estimated average shoreline retreat at each sector is 1.1, 4.0, and 1.9 m/y at SBN and SBM, MSM and MS1, and MS2, respectively (see Figure 1 for locations). It is assumed that current evolution trends remain constant during the timeframe of the analysis, which is limited to 20 years. However, this could be substituted by time-varying evolution rates provided this should be the case.

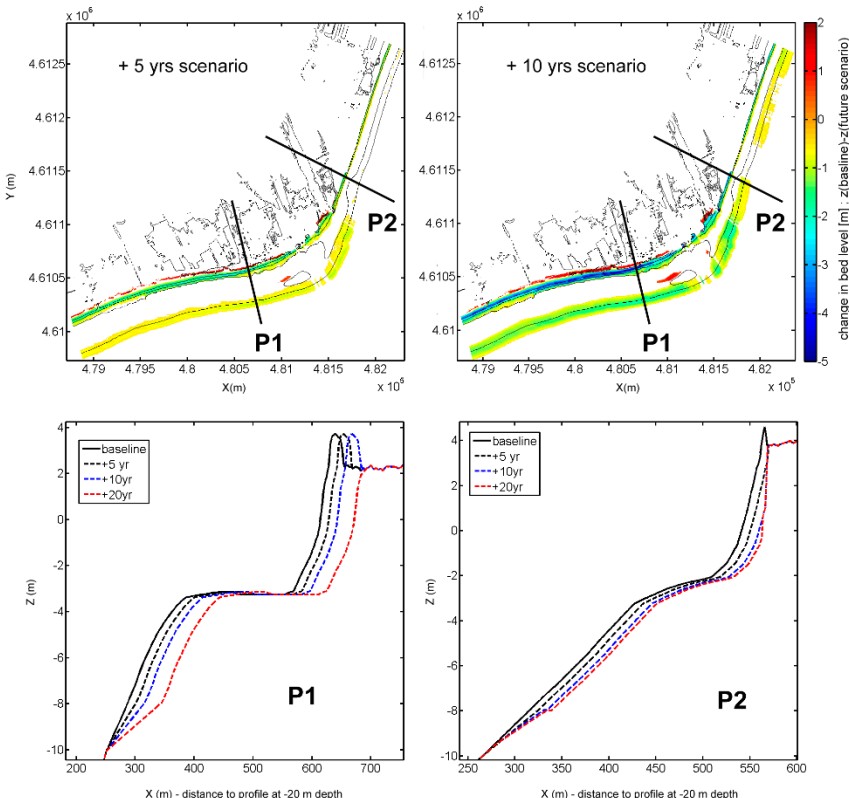


**Figure 5: Changes in the bed level grid for the future scenarios. Difference between baseline bed level and scenario bed level (upper). Profile retreat at both sides of the river mouth at the different time horizons (lower). P1 belongs to MS1 (retreat of 4 m/yr) and P2 belongs to SBM (retreat of 1.1 m/yr).**

Thus, to account for this background response, each scenario was defined based on a given coastal morphology at a given time horizon. The baseline morphology, which corresponds to the current scenario, is the one described in Section 2 (Figure 1) that was directly measured. Future coastal morphology for each scenario corresponding to different time horizons (+5 years; +10 years; and +20 years) were built by retreating the active part of the shoreface, from a -10 m-depth to the subaerial beach, according to erosion rates at the different areas. This hypothesis about the shape of long-term (decadal) profile

changes follows the hypothesis applied in shoreline evolution models, i.e. a parallel displacement of the active profile from the emerged beach down to the depth of closure (e.g. Hanson, 1989). To ensure alongshore smoothness after retreating, linear transitions between sectors affected by different retreat rates were applied. Resulting configurations for two scenarios are shown in Figure 5, along with example profiles at locations under different levels of background retreat. Local constraints due to the lack of accommodation space due to the existence of hard structures at the hinterland were also

considered. When the shoreline reaches a fixed structure limiting the landward translation, it is assumed that, locally, the beach disappears and, in consequence, no further profile retreat will occur. As an example, Figure 5 shows the beach profile retreat at two locations with different hinterland characteristics: P1 has no hard limit, whereas P2 is limited at the back by a promenade. This results in a continuous retreat of P1 for all scenarios, whereas the retreat of P2 is limited at the beach after 10 years.

**3.6 Bayesian Network integration**

The BNs are probabilistic models based on acyclic graph theory and Bayes theorem (Pearl, 1988; Jensen, 1996). They have demonstrated their versatility and utility in efficiently combining multiple variables to predict system behaviour. Within the context of this work, they can be used to represent the SPRC scheme through the dependency relations between the different steps (see e.g. Straub 2005; Jäger et al. 2018). In this sense, they can easily be adapted to assess different natural hazards and

their impacts on many kinds of receptors, for both descriptive as well as predictive applications (see e.g. Beuzen et al. 2018b).

In this work, two BN configurations were used to characterise the system response to the impact of coastal storm events. This was done to optimise the BN structure by limiting the number of variables per network while solving the different parts of the SPRC framework. In practice, one BN solved the source-consequences relationships (BN-A), while the other

characterised the receptor-consequence spatial distribution (BN-B), providing complementary information on the local risk profile.

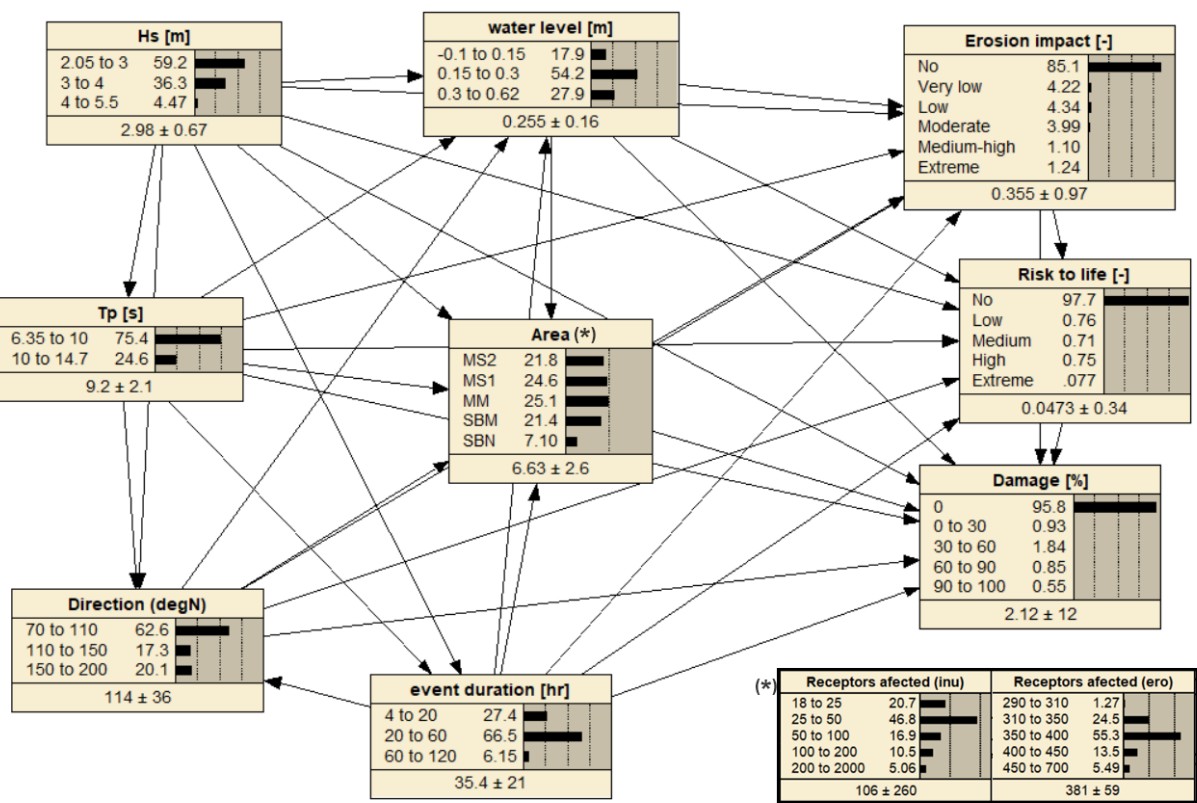

**Figure 6: BN-A, linking source variables to consequences. Central variable (*) is used for conditioned assessments and is one of**
**three: (i) Total number of affected receptors by inundation within a storm event; (ii) total number of affected receptors by erosion**
**within a storm event, and (iii) receptor area (i.e, SBN, SBM, MSM, MS1, and MS2). Distributions correspond to the baseline**
**scenario.**

BN-A (Figure 6) links storm-defining variables (Hs, Tp, duration, direction, and water level) and impacts to the receptors (erosion impact, risk to life, and structural relative damage). The central variable of the network (indicated by * in Figure 6)

was used to perform conditioned assessments. Depending on the objective of the analysis, it can be (i) the total number of affected receptors by inundation within a storm event; (ii) total number of affected receptors by erosion within a storm event, or (iii) receptor area (SBN, SBM, MSM, MS1, and MS2), as shown in Figure 6. To account for the spatial extension of the impacts, we included the total number of affected receptors as an output variable. These are counted outside the BN for each simulated storm peak and introduced in the BN as an additional storm characteristic variable. To characterise the extension

of inundation, all receptors presenting a relative damage other than 0%, or a risk to life other than "None" were counted. Similarly, to characterise the extension of erosion, all receptors presenting an impact level different than "None" were counted. In practical terms, this means that, in general, the number of affected receptors by erosion was larger than by inundation. This is because, with the used criteria, it is quite probable to have receptors affected by "Very Low" to "Moderate" erosion risks representing the loss of protection provided by the beach, although this does not imply that they

will be directly exposed to wave impact. However, inundation-related impacts are always associated with the presence of water at the receptors. This has to be taken into consideration when interpreting the obtained results.

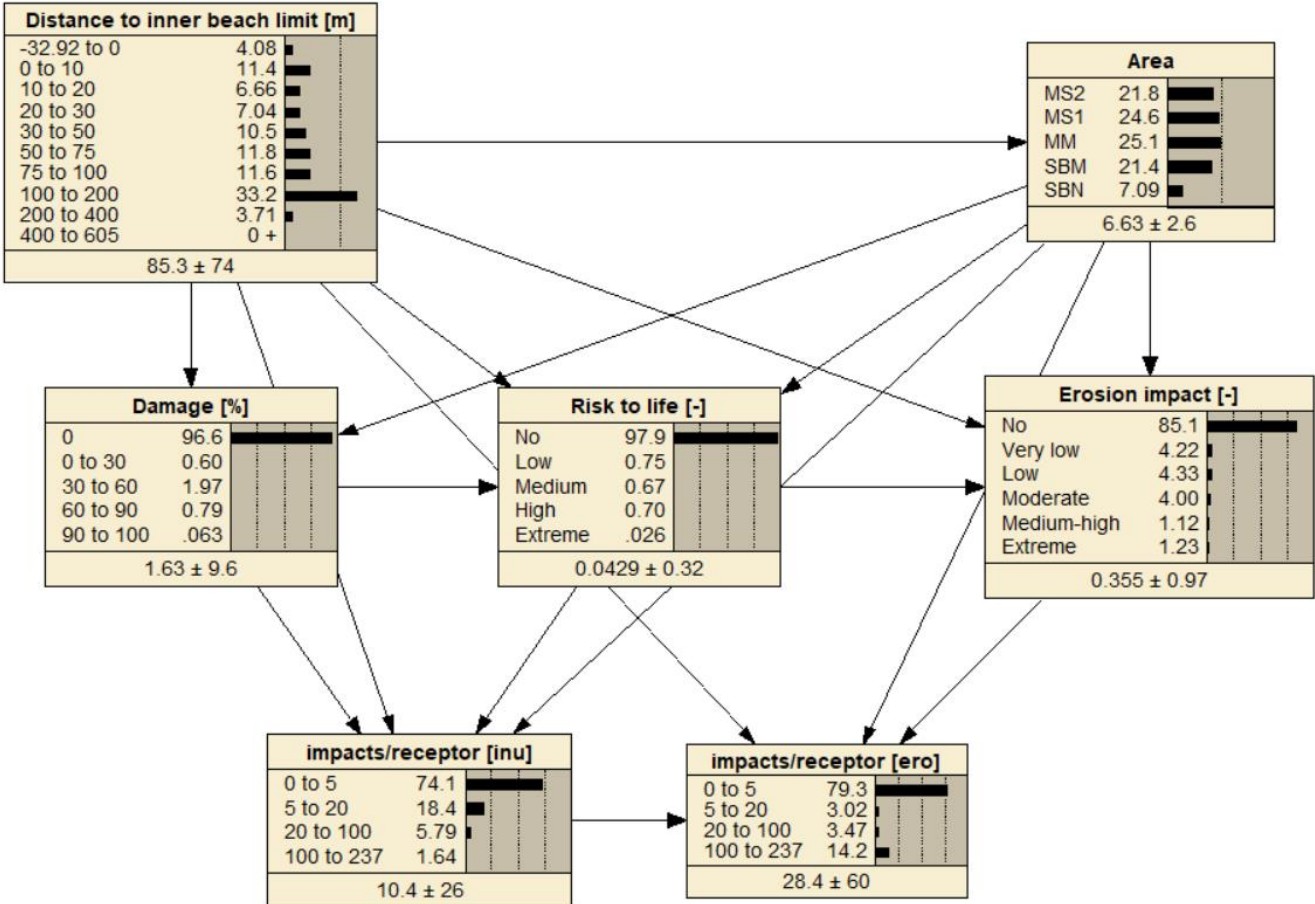

**Figure 7: BN-B, linking consequences to receptors spatial locations. Distributions correspond to the baseline scenario.**

BN-B (Figure 7) links the simulated impacts on the receptors to their position, characterised by their location along the coast (area) and the distance to the inner beach limit.. These variables provided additional insight into the system response, as the obtained distributions with the BN merge storm-climate variability and the spatial distribution of receptors. For the inundation risk, the number of impacts with damage different from 0%, and/or with risk to life different from "None" was counted at each receptor. For the erosion risk, the number of impacts different from "None" was counted per receptor. This

has the same consequence as that described in the previous case (BN-A) for interpreting the obtained results. It must be noted that from all receptors displayed in Figures 1d, 3, and 4, only those presenting at least one impact for the entire storm dataset, by either inundation or erosion, were used for the BN training. Otherwise, the choice of receptor population to include in the assessment would be arbitrary, affecting the obtained distributions.

The presented BN-model was designed to assess storm-induced risks in a coastal hotspot where the storm climate and coastal response are well known (e.g. Jiménez et al. 2018; Sanuy et al. 2020). Due to this, the discretisation of variables (Figures 6 and 7) was done manually, enabling better accuracy than automatic unsupervised methods and closer accuracy to supervised discretisation with less associated variability on model performance (Beuzen et al. 2018a). Notably, both BNs present a certain degree of complexity given the discretization level of some variables and the number of variables used. The BNs are

designed to be descriptive BNs (Beuzen et al., 2018b), and thus, source variables are also interconnected to avoid the propagation of noise from empty combinations to the output. This departs from predictive BNs which aim to infer system behaviour and predict combinations beyond those learned from the dataset.

## 4. Results

Figures 3 and 4 show the results of a single simulation exercise for 1 of the 455 possible events. Each simulation results in the collection of the BN variables characterising the storm characteristics together with the location and the risk values for each receptor (~4000). The following subsections present the results of the integration of multiple simulations (i.e. 179 in baseline morphology and 69 for each additional scenario). First, the 69-storm *subset* is validated against the 179-storm original dataset using the baseline morphology to ensure that it properly represents the local storm climate. This is followed by the presentation of the risk characterisation of the Tordera Delta, starting with risk probabilities integrating all storms and receptors (global risk probabilities), and then, with conditioned probabilities between forcing-area risk (BN-A, Figure 6) and area-distance risk (BN-B, Figure 7).

### 4.1 Subset validation

Table 5 shows the obtained statistics using Eq. 1 and 2 to compare the discrete probability distributions obtained with the BN using the 179-storm dataset against those from the 69-storm *subset*.

All obtained values of the mean significance $\bar{S}$ and its root mean square (*RMS*) are close to 0; therefore, from the perspective of obtained results, it can be assumed that the obtained distributions by feeding the BNs with the subset almost identically represent the same source population as that of the complete dataset. This is true both for global distributions and for conditioned discrete probability density functions (PDFs).

**Table 5: Results of the histogram comparison between the original storm dataset and the *subset* for the baseline scenario.**

| Verification case | $\bar{S}$ | RMS |
|---|---|---|
| **Global risk probabilities** | | |
| Histograms of *Damage*, *Risk to life* and *erosion impact* variables without conditioning (Fig. 6 and 7) | -0.009±0.006 | 0.04±0.05 |
| **Risk probabilities conditioned to source characteristics** | | |
| *Hs, duration, water level, and direction* conditioned to *Damage*, *Risk to life* and *erosion impact* levels at different *areas* | 0.0006±0.02 | 0.05±0.03 |
| **Risk probabilities conditioned to receptors locations** | | |
| *Damage*, *Risk to life* and *erosion impact* probabilities at the different *areas* and *distance to the beach* (Figures 10 to 12) | 0.0041±0.02 | 0.04±0.08 |

### 4.2 Risk characterisation

Table 6 shows the obtained probability levels for different tested scenarios in the study area. These so-called prior (unconstrained) probabilities represent the expected frequency of the different risk levels in the study area and account for the variability of the source (storm climate), spatial distribution, and extent of the impacts on the receptors. In general, under current conditions, the probability of receptors being affected by significant (high and extreme) risks is low (1–2%). However, the existence of background erosion in the study area results in a significant increase in future risks. Under the baseline scenario, the computed probability of moderate-high risks associated with erosion is larger than the ones for inundation. However, when we only consider those cases where erosion results in exposing receptors to direct impact (high and extreme risk), the obtained probability values are of the same order of magnitude as those obtained for moderate damages associated with inundation. Additionally, results of number of affected receptors from BN-A (not shown in the table) show an increase in the % of storm conditions affecting a large number of receptors along the study area. As an example, storm conditions with the potential to affect more than 200 receptors with any level of inundation risk increases from 4% under current conditions to 20% and 40% after 10 and 20 years, respectively. Simultaneously, storm conditions

affecting more than 450 receptors with any level of erosion risk will rocket from the current 4% to 100% in 10 years. Here, it is important to remember that erosion risk is not only related to direct impact but also the loss of protection function (decrease of beach width in front of a given receptor), while inundation risk implies the direct effect of water on the receptor. In general, estimated probabilities associated with erosion-induced risks are larger than those due to inundation when comparing similar risk levels.


Figure 8 shows the alongshore-spatial distribution of the BN-computed percentages of receptors affected by any level of risk induced by both hazards under all scenarios. Obtained results show a different spatial behaviour according to the considered hazard. Thus, the most erosion-affected areas (those showing a larger percentage of receptors with damage different to zero) are located northwards of the river mouth, whereas areas southwards of the river mouth are more affected by inundation (higher probability values). The time evolution of the affected receptors is also different, reflecting existing spatial variations in shoreline retreat rates. Thus, the largest relative increase in the number of impacted receptors under future scenarios occurs southwards of the river mouth. Notably, the MS2 sector is the most sensitive to future risks, as currently, although it is well protected by a relatively wide beach, this protection will fade after 10 to 20 years.

BN-A was also used to characterise the conditioned probabilities of storm characteristics associated with the highest risks and assess whether these probabilities vary along the study area. As seen in Figure 9, under current conditions, the main storms driving the highest inundation-induced risks are characterised by Hs higher than 4 m and from the E direction. This is valid for the entire area, although their relevance slightly varies along the coast. Thus, the only exception is found in the SBN sector, where the promenade is so close to the shoreline that lower Hs can induce inundation damages. For future conditions (20 years scenario), the relative importance of storms with smaller Hs increases, and the relative importance of present secondary wave directions, S and SE, also increases in relative terms.

Table 6: Global risk probabilities for different risk levels under the different scenarios. Note that global risk probabilities account for the variability in the source (storm climate) and the spatial distribution of impacts on the receptors.

| Global risk probabilities | Baseline | + 5 yrs | +10 yrs | + 20 yrs |
|---|---|---|---|---|
| **Inundation** | | | | |
| Moderate risk or higher (damages ≥ 30%) | 3% | 5% | 5% | 7% |
| Moderate risk to life or higher | 2% | 3% | 3% | 5% |
| High and extreme risk to life | 1% | 2% | 2% | 3% |
| **Erosion** | | | | |
| Moderate risk or higher | 6% | 9% | 13% | 13% |
| High and extreme risks | 2% | 4% | 8% | 8% |

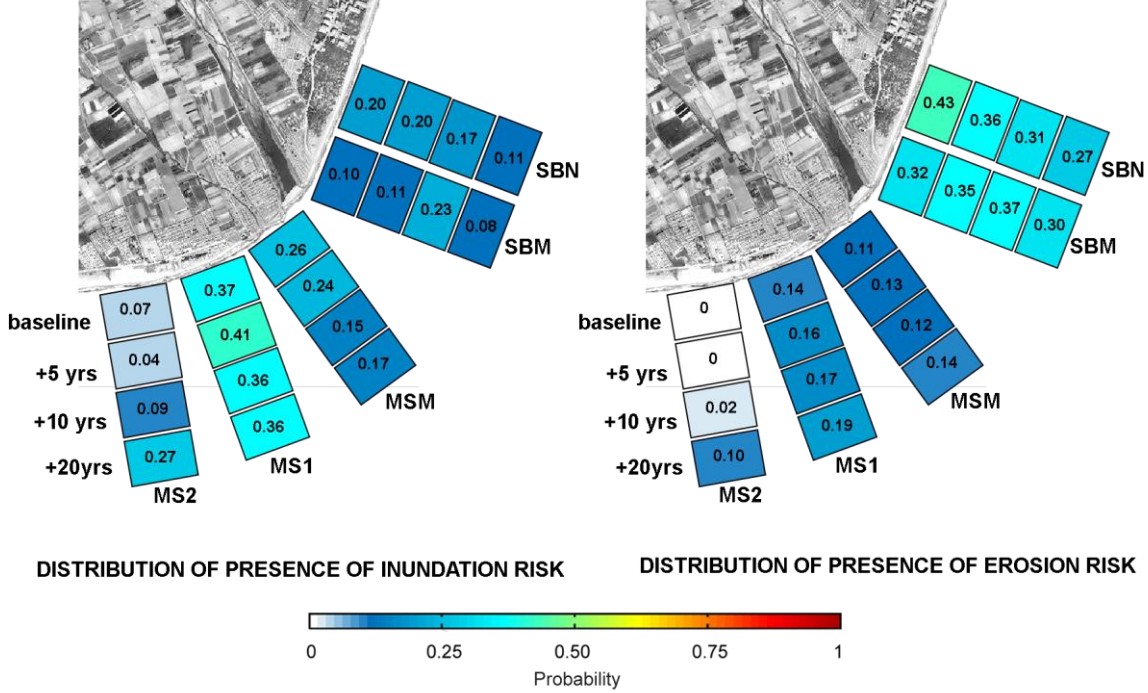

DISTRIBUTION OF PRESENCE OF INUNDATION RISK    DISTRIBUTION OF PRESENCE OF EROSION RISK


**Figure 8: Distribution of risks (at any level) across the different sectors (see specific locations in Figure 1). This shows the relative proportion of impacted receptors in the different areas, under the baseline morphology and the future +5, +10, and +20 yr scenarios. Orthophoto provided by Institud Cartogràgic i Geològic de Catalunay (ICGC).**

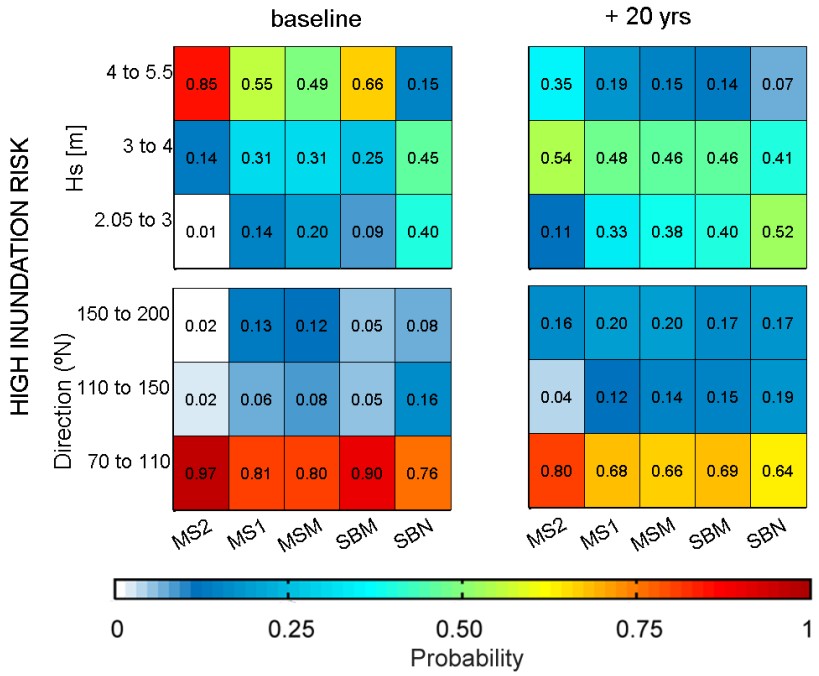

**Figure 9: Probability of storm Hs and direction conditioned to the area and to highest intensity inundation risk, i.e. moderate to high risk to life together with high structural damages (≥ 60%). Note that extreme risk to life and damages over 90% are not present for the study site. Results must be read as individual vertical histograms (1 histogram per area).**

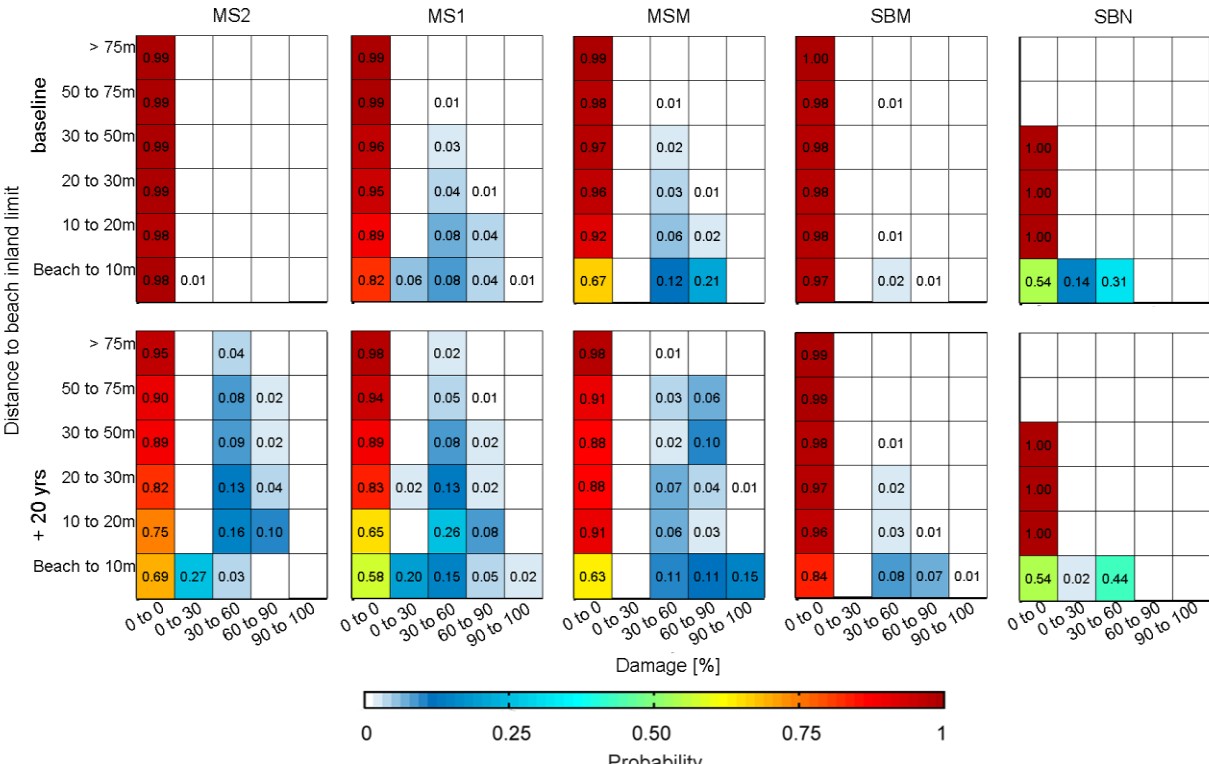

**Figure 10: Probability distributions of the relative damage by inundation conditioned to the different subareas (see Figure 1 for locations) and the distance to the inner limit of the beach. Baseline and +20-year time horizon of background shoreline retreat. Results must be read horizontally as individual histograms for each combination of area, distance, and scenario.**

The spatial distribution of the expected impacts across the study area was analysed using the BN-B. The objective of the analysis was to assess the probability damage occurring at the receptors located at a given distance from the beach (i.e. limit between beach and hinterland). Figures 10 and 11 show obtained results in terms of % of inundation-induced damage and risk to life, respectively, for different time horizons. Consistent with the results shown in Table 6, under current conditions (baseline), storms cannot induce extreme structural damage (>90%) (Figure 10) nor extreme risk to life (Figure 11). High damages (> 60%) are mainly concentrated at the outer fringe of the hinterland of the two locations (MSM and MS1) with associated conditioned probabilities of 21% and 5%, respectively. These two areas also show the highest probabilities of risk penetration into the hinterland. Northwards of the river mouth, the SBN sector presents a large probability of moderate damages, but it is limited to the external fringe. Regarding risk to life, a similar spatial pattern is observed, with MSM showing the largest probability of high risk (20%) at the external fringe, SBN at the north with 12%, and MS1 only showing a residual 3%. The obtained results reflect the role played by the current coastal morphology, where the southern area is characterised by narrow and low elevation beaches (MSM and MS1), whereas the SBN sector in the north is composed of a narrow beach backed by a promenade. Notably, SBM with a narrow beach but higher topography without a promenade and MS2 with low topography but wider beaches are the areas presenting the lowest risks.

Under future conditions (+20-year scenario), significant changes are observed in the intensity of risks and extension across the territory (Figures 10 and 11). The spatial modulation on induced risks as a consequence of the beach narrowing due to background erosion is especially evident in the southernmost area, MS2. Whereas this sector does not experience any risk under current conditions, significant probabilities of moderate and high damage and risk to life is expected to occur in 20 years, not only at the outer fringe but also in inner positions of the hinterland. The other sectors along the coast also show significant increases in the probability of occurrence of any type of risk and extension of the impacts landward (Figures 10 and 11).

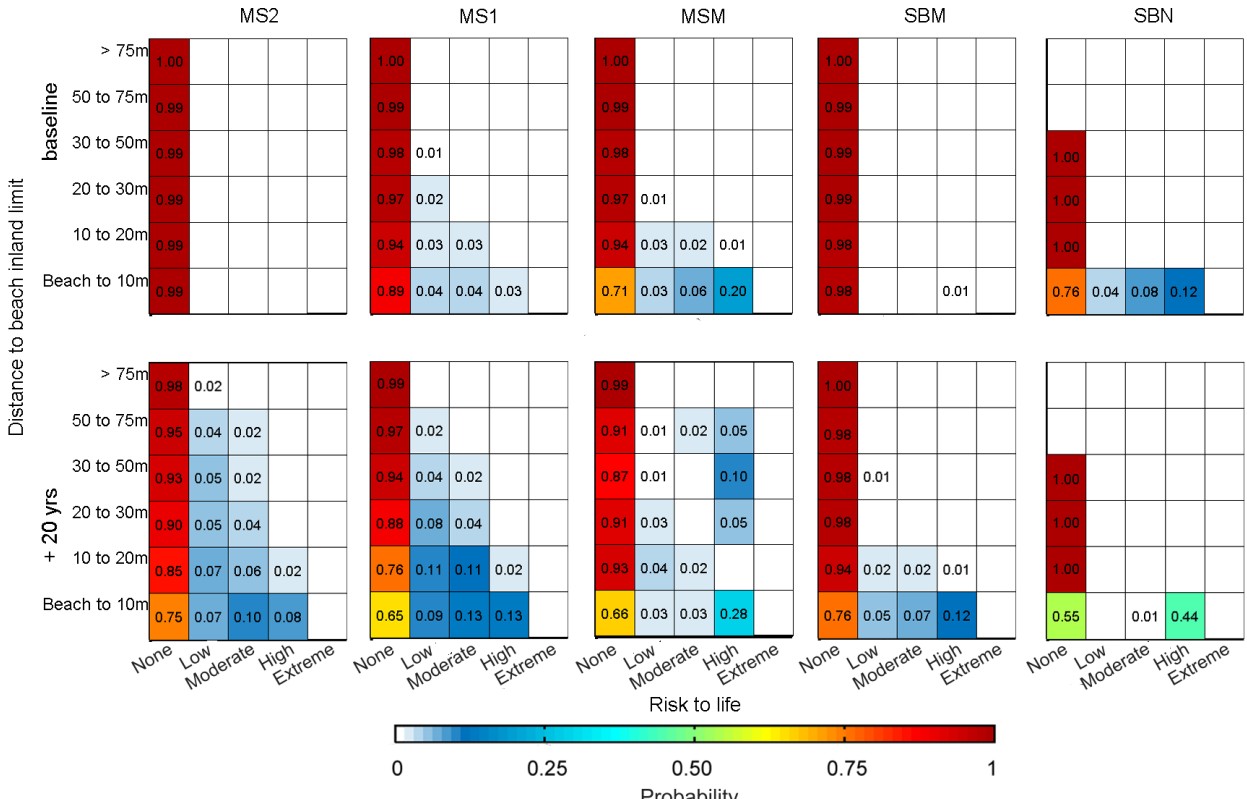


**Figure 11: Probability distributions of the risk to life by inundation conditioned to different subareas (see Figure 1 or 8 for locations) and the distance to the inner limit of the beach. Baseline and +20-year time horizon of background shoreline retreat. Results must be read horizontally as individual histograms for each combination of area, distance, and scenario.**

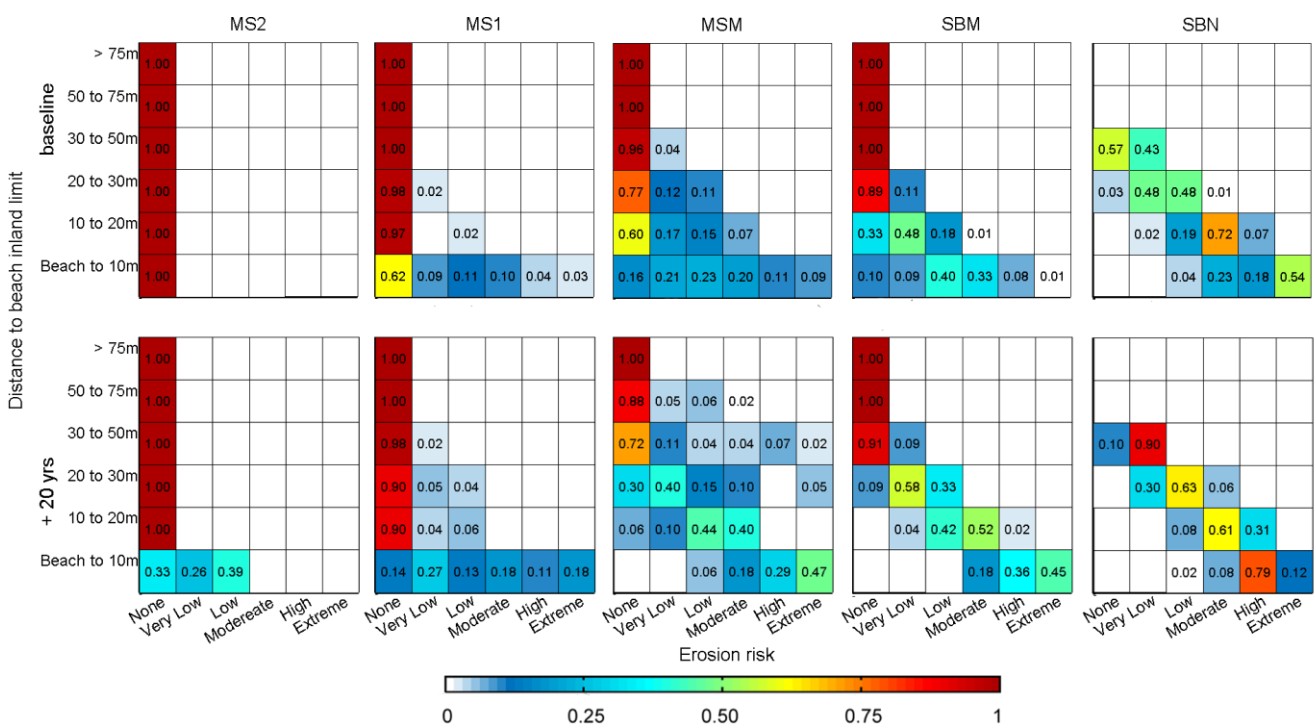

**Figure 12. Probability distributions of the erosion risk conditioned to the different subareas (see Figure 7.1 for locations) and the distance to the inner limit of the beach. Baseline and +20-year time horizon of background shoreline retreat. Results must be read horizontally as individual histograms for each combination of area, distance, and scenario.**

The spatial distribution of erosion-induced risk under current conditions (Figure 12) reflects the existence of hard elements

and varying beach widths along the study area. Thus, SBN presents the largest probability of extreme risks at the promenade (54%), followed by MSM (9%), and MS1 (3%). In SBN, the promenade acts as a physical boundary for erosion; the distribution of risk levels into the hinterland shows a linear pattern reflecting its position. At the southernmost end, MS2 is

currently well protected by a wide beach and no risk is predicted under the current conditions. Under the +20-year scenario, the effect of the promenade in SBN is reflected through the unaltered spatial pattern of affected locations and computed

probabilities. MSM and SBM show the largest relative increase of extreme risks (probabilities of 45% and 47%, respectively) at receptors located closest to the beach, along with the largest spatial propagation of risks into the hinterland, as no hard elements are present to limit the retreat of the shoreline. At MS1, the probability of extreme risks increases to 18% at the beach limit with small changes at larger distances, while MS2 starts presenting significant probabilities of low risks indicating that the beach will begin to decrease its protective function against storm impacts after 20 years.

**5. Discussion**

In contrast to previous applications of the BN-SPRC concept presented in Jagër et al. 2018 (e.g. Van Verseveld et al., 2015; Plomaritis et al., 2018; Ferreira et al. 2019; Sanuy et al., 2018), this paper presents a fully probabilistic characterisation of the source using all available storms in a 60-year long wave time-series hindcast, following the response approach, and modelling their induced erosion and inundation risks over all the identified receptors at the study site.

The methodology was successful in identifying storm characteristics with higher probabilities to induce given risk levels for different coastal hazards (inundation and erosion). It was efficient in assessing the expected changes in storm characteristics and probabilities under different scenarios, which were developed based on the background mid-term coastal evolution. In this sense, the obtained relationship under current conditions of erosion and inundation risks with storm direction and Hs depicts the general characteristics of storm-induced hazards in the study area (Mendoza et al. 2011). The thresholds used to

identify independent events in the P.O.T are site dependent. In this work, they agree with the storm classification in Mendoza et al., (2011), and therefore they are valid for the Catalan coast (NW Mediterranean). The BN output showed a lack of correlation between high risks and water levels, consistent with the previous findings of Mendoza and Jiménez (2008) on the non-relevance of storm surges. Under future conditions, the background shoreline erosion changes the sensitivity of the area to storms. Thus, for the tested scenarios, the population of storms with potential to significantly impact the area

increases and higher risks will be associated with storms characterised by lower Hs with currently secondary wave directions (Figure 8). If we combine this larger exposure to southern storms with the large sensitivity of the area to the impact of such S storms (Sanuy and Jimenez, 2019), this may have serious implications for the future risk management of the area.

The method has been designed to provide a detailed spatial assessment to assess the sensitivity of the area, which permits the association of the local risk profile with different morphological characteristics such as beach orientations, height, and the

presence of hard structures. In this sense, the local response affected by the presence of the promenade at S'Abanell (SBN), and revetment in Malgrat North (MS1) were adequately characterised by the BN. This spatial analysis also permitted the assessment of a differentiated variation in future risks along the study area. Thus, whereas some areas being currently exposed linearly increased the probabilities of higher risks, other areas currently well protected will be subjected to higher future risks without any variation in storminess.

The method can also be used for testing risk management measures such as the performance assessment of different setbacks. While this measure is effective in reducing coastal damages in eroding coastlines, especially in the context of climate change (Sanó et al. 2011), it has to be defined for given time horizons and driving conditions (e.g. Wainwright et al. 2014). To this end, the framework presented herein permits the definition of probabilistic setbacks at the study site. Moreover, as this definition is based on the probabilistic distributions of the different risk levels and impacts per receptor at

different locations across the coastal domain, it differs from existing approaches that are essentially based on the probabilistic definition of the shoreline position (e.g. Jongejan et al. 2016). As an example, Table 7 shows the calculated minimum distances landward of the inner limit of the beach according to different risk levels for different time horizons (scenarios). As the BN output combines the natural variability of the storm climate with the spatial variability of the

impacted receptors, setbacks can be defined from these (total probability, as in Figures 10 to 12) or by assuming that the presence of a given risk level must be completely tackled, focusing then only on the spatial distribution of receptors under such levels. The second approach will result in more conservative (wider) buffers. Table 7 shows the calculated buffer distances using both perspectives. The obtained setbacks accounting for the total probabilities can be used as proposals for managed retreats, as they reflect the areas with a high number of impacts per receptor; the setbacks defined by the presence of a given risk level can be used to inform self-preparedness against risk, as they highlight zones where the existence of risk is possible but highly infrequent. It must be noted that all scenarios have been simulated without any assumption of receptor re-allocation, and therefore, hard limits for erosion remain homogeneous across scenarios. Therefore, the distances presented in Table 7 must be interpreted as the evolution of the baseline setbacks at different horizons in a business-as-usual situation.

**Table 7: Characterisation of setbacks for different hazards and risk levels in the Tordera Delta. Baseline scenario and +20-year time horizon using two approaches: (i) Total probability, i.e. natural variability of the storm climate with the spatial variability of the impacts on receptors and (ii) risk presence, i.e. focusing only on the spatial distribution of receptors under that level.**

| Area | Setbacks (m) | | | | |
|---|---|---|---|---|---|
| | Moderate inundation damage (>30%) | Moderate Risk to Life | High Risk to Life | Low Erosion Risk | High and Extreme Erosion Risks |
| **Baseline - based on total probability** | | | | | |
| **MS1** | 10 | 10 | 0 | 10 | 5 |
| **MSM** | 10 | 10 | 10 | 30 | 10 |
| **SBM** | 0 | 0 | 0 | 25 | 8 |
| **SBN** | 10 | 10 | 10 | 50 | 15 |
| **Baseline - based on risk presence** | | | | | |
| **MS1** | 98 | 43 | 9 | 8 | 7 |
| **MSM** | 196 | 110 | 19 | 38 | 9 |
| **SBM** | 150 | 71 | 41 | 23 | 9 |
| **SBN** | 10 | 10 | 10 | 44 | 16 |
| **+ 20 years - based on total probability** | | | | | |
| **MS1** | 50 | 20 | 10 | 25 | 10 |
| **MSM** | 55 | 50 | 10 | 75 | 50 |
| **SBM** | 10 | 10 | 10 | 40 | 10 |
| **SBN** | 10 | 10 | 10 | 50 | 20 |
| **+ 20 years - based on risk presence** | | | | | |
| **MS1** | 137 | 49 | 10 | 24 | 5 |
| **MSM** | 130 | 98 | 71 | 69 | 44 |
| **SBM** | 111 | 109 | 29 | 38 | 10 |
| **SBN** | 10 | 10 | 10 | 47 | 18 |

The presented method is based on the response approach (Garrity et al., 2006; Sanuy et al., 2020a) as it produces probabilities based on how hazards (erosion and inundation) affect the receptors in each of the storm events derived from a long dataset of 60 years; it does not allow the extrapolation of the storm conditions out of the range of the ones registered in such datasets. This has relatively less impacts on the results when compared to the impacts from other sources of uncertainty, such as morphological variability or model error (Sanuy et al., 2020b). Nonetheless, it allows the simulation of all storm events with their real shapes (time evolution of storm characteristics) without introducing large uncertainty in hazard estimation associated with the use of synthetic storms that are commonly used to define the shape of statistically extrapolated storm events (see e.g. Duo et al., n.d.).

In this study, hazards were computed using a robust model to simulate the storm-induced coastal response, XBeach, calibrated for an event representative of extreme conditions (see Sanuy and Jiménez, 2019). They were converted to risk by using damage curves recommended for use in the study area. However, the BN methodology is flexible for any kind of

model, as well as to include model uncertainties (using different models or setups) and measurements (e.g. Sanuy et al., 2020b for cross-shore parametric models) to extend the data training and improve the results while testing its predictive capacity.

With regard to building future scenarios to assess future risks, we have limited the present study to mid-term scenarios, i.e. at the decadal scale (20 years). They were built based on decadal-scale shoreline rates of displacement retreat measured by Jiménez and Valdemoro (2019), which were used to build future coastal configuration assuming that no changes in evolutive conditions will occur. Even in this case where no changes in forcing conditions were applied (no changes in storm conditions nor sea level rise), this approach permitted the identification of significant changes in the storm-induced risk profile.

It has to be mentioned that to build these morphological scenarios, it is necessary to "forecast" future configurations of the shallow water bathymetry. In this work, this was done by extending shoreline displacements down to the depth of closure by assuming a simple parallel displacement of the active inner profile, which is compatible with the usual hypothesis applied in mid-term shoreline models. However, other profile change modes could also be applied, such as a wedged-shaped change over the closure depth to simulate a slower retreat of the delta front in comparison with faster shoreline changes (e.g. Refaat and Tsuchiya, 1991). In both cases, their morphological consequences are limited to the shallowest and faster part of the shoreface and, in consequence, are strictly applicable to expected mid-term (decadal) changes. Building longer-term morphological scenarios would require to consider other options since the depth limiting significant changes in the beach profile will extend further with time scale (e.g. Cowell et al. 1999 ). In this line, Stive and de Vriend (1995) proposed a long-term shoreface evolution model that considers a varying type of change through the shoreface, from an upper part experiencing a parallel displacement, to a declining/inclining lower shoreface down to the inner shelf limit.

In the case of structures/barriers being exposed at the shoreline along the study area due to background erosion, we have assumed that, locally, the active profile will not retreat further once the beach had disappeared. In the event of such situation, the structure would be subjected to the highest possible risk and as so would be classified in the framework. Further bottom variations in front of the structure which may lead to its collapse due to scouring will not modify this classification.

In any case, it has to be considered that building future morphological scenarios to forecast the evolution of coastal risks at long-term scales will add uncertainty to the analysis, in addition to that associated with expected varying climatic forcing, since long-term morphodynamic modelling integrating all relevant processes is still an unsolved issue (e.g. Ranasinghe, 2020).

## 6 Summary and Conclusions

Bayesian networks have proven to be an efficient tool to develop an SPRC-based framework for probabilistic storm-induced risk assessment and risk mapping at a local scale (few kilometres). In this work, BN training has been carried out using storm events identified in a 60-year long wave time-series, and simulated hazards and corresponding risks were evaluated at the receptor scale (few metres). This resulted in a full representation of the storm climate (source) leading to probabilistic characterisation of risks that accounted for climate (storms) and geographic (receptor location) related variabilities, as the BN training followed the response approach (i.e. the simulation of the coastal response for all identified storms). The framework is also able to predict how risks will evolve in the near future, both in intensity and spatial distribution, provided that climate and/or geomorphology scenarios are built. One of the advantages of the framework is that it permits the identification of conditional probabilities, and thus, the identification of which are the storm characteristics that induce risks of a given magnitude. This is a very useful property in designing disaster risk reduction (DRR) strategies and measures including the design of early warning systems.

Concerning the analysed case study, the Tordera Delta (NW Mediterranean coast) presents, under current conditions, a larger susceptibility to storm-induced erosion than to inundation, which was identified through computed probabilities of high-risk

levels associated to both hazards along the coast. Storms inducing the largest impacts are characterised by high Hs (>4 m) for inundation and long duration (>60 hours) for erosion. In both cases, these correspond to Eastern events, which are the most energetic in the area.

The application of the framework for future scenarios predicted an increase in the local risk as a larger number of storms will be able to induce higher risk levels. As these scenarios were built by projecting the coastal configuration up to two decades

from now (based on background erosion), the framework reflected the morphodynamic feedback resulting from the loss of protection provided by progressively narrowing beaches. In addition to the increase in risk levels, it also identified a change in storm threshold conditions affecting the area in a significant manner, characterised by lower Hs values and with an increasing importance of southern events.

Finally, the obtained spatial distribution of risks permitted the identification of the most sensitive areas and their evolution

over time. This can be used to make decisions on the required DRR measures both along the coast and across the hinterland. The use of the BN to obtain probability distributions of the different risk levels across the hinterland allowed for a probabilistic definition of setbacks.

**Competing interests.** The authors declare that they have no conflict of interest.

**Author contribution. M. Sanuy**: Conceptualization, Methodology, Software, Formal analysis, Writing - original draft, Writing - review & editing. **JA Jiménez:** Conceptualization, Resources, Writing - original draft, Writing - review & editing, Supervision, Project administration, Funding acquisition.

**Acknowledgements.** This study was conducted in the framework of the RISC-KIT (Grant No 603458) and the *M-CostAdapt* (CTM2017-83655-C2-1-R) research projects, funded by the EU and the Spanish Ministry of Economy and Competitiveness (MINECO/AEI/FEDER, UE) respectively. The first author was supported by a PhD grant from the Spanish Ministry of Education, Culture and Sport. The authors express their gratitude to IH-Cantabria for supplying wave and water level data, to the Spanish Ministry for Ecological Transition for the bathymetric data, and to the Institut Cartogràfic i Geològic de

Catalunya by for the LIDAR data used in this study.

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
