# Peer review of "Probabilistic characterisation of coastal storm-induced risks using Bayesian Networks"

_Natural Hazards and Earth System Sciences, 2020_

## Referee Comment (RC1) · Anonymous Referee #1 · 28 Aug 2020

I have reviewed the manuscript entitles 'Probabilistic characterisation of coastal storm-induced risks using Bayesian Networks' by Sunay and Jimenez.

Overall the article is of high quality and provides an alternative method for using BN in risk assessment that although it is based on the source-pathway-receptor-consequences concept it has some novel methods related with the storm selection. I believe that the article is of high interest for the journal and well within the journals scope. However, I believe that in order for the manuscript to be accepted some changes need to be addressed more for clarifying some aspects of the work and for providing further information and limitations of the method.

[Figure]

General comments:

The Abstract of the article although correct is rather general and it is not highlighting the results and the novelty of the work. I believe some addition of more specific results that are present in the discussion will benefit the current version of the abstract. Most of my comments are concentrated in the methodology sections this is partly because the method is rather complex and the proposed novelty although important it is not obvious from the begining. The results and discussion sections are very well written and explained with high quality figures that although sometime complex they concentrate a large amount of information.

I have made specific comments in the text where I have questions or doubts my maid concerns at the moment is that novelty of the method is not properly described in risk terms. I believe that the BN approach proposed is valid for characterizing the risk for the entire storm climate and not for specific storms as proposed by previous works. However if this is true it needs to be highlighted by the authors in the abstract and in the title is necessary. My secondly concern is related with the scenarios proposed. Some more explanation is needed on why the shoreline retreat is extended to the entire shoreface.

Specific comments:

LINE 33: source terms are booth the storms and the storm induced hazards.

LINES 53-58: Plomaritis et al 2018 select the events using the same methods as Poehekke et al., 2016. The method is based on a series of colula applications using Hs as a main parameters. I don't think that this method can be consider non-probabilistic but indeed the method can differ. Please explain with more detail the differences in the storm selection. Poehekke et al., 2016 also follows the ideas of response approach with the use of copulas but with triangular storms. I believe that the discussion over the different approaches that the authors provide is very interesting and I would suggest extending it or order for the reader to be better informed on the sometime small but

important details. The reference Duo et al., needs updating.

Study area: Provide the names of the areas in Figure 1 not only the code. Now they is given in Discussion but the codes are used before. I think some information of the areas and the logic behind the separation could be interesting.

LINE 95: I think the paper Sanuy et al. (2018) is not in the reference list.

LINE 143: Provide the number or persetnage of empty groups LINES 174-175: How many storms per bin you have in the subset group and which are the output paramters you test? My understanding so far is that you have one storm per group in the subset so, I am not sure how you calculate the variance per bin. Are you evaluate the BN output or input with the equations 1 and 2 or the entire BN?

Hazard Assessment: Which are the indicator (model output parameter) you use for each hazard.

LINES 194-198: The area characteristics can be put in the study area. See my previous comment.

LINES 246-248: Given the steep slopes of the study area I understand the extrapolation of the shoreline retreat values to the upper beach (-2 to -4 m) but continues retreat up to -8 suggest a huge amount of sediment loss and that all sediment from the upper beach is removed by longshore drift. I am not an expert on Catalan coast but some additional justification for the selected scenarios must be provided.

LINE 272: Why the storm parameters are linked in Figure 6? How is te term of previous energy is incorporated in the BN?

LINES 274-277: The central variables i and ii are not shown in Figure 6. Please provide more details. Explane where the estimation of the total number of receptors is done, in the BN or before?

LINES 420-421: What are the advantages of this fully probabilistic BN? I suppose that

the previous papers were focused on the individual storm assessment while here is attempted an integrated assessment of the storm conditions. If this is correct it has to be stated and event introduced in the abstract.

---

## Referee Comment (RC2) · Anonymous Referee #2 · 7 Oct 2020

I have reviewed the manuscript entitles 'Probabilistic characterisation of coastal storm-induced risks using Bayesian Networks' by Sunay and Jimenez.

Overall the article is very well written and of high quality. It presents a new framework/approach using the SPRC framework to examine coastal vulnerability to erosion and inundation at an area within the Spanish coastline exposed to Mediterranean storms. The methodology uses Bayesian Networks to take the SPRC inputs/outputs to create a probabilistic outcome of risk assessment. I believe that the article is well within the journals scope and will be of interest to the readers. However, I believe some changes are needed and points clarified as detailed below.

[Figure]

General comments: Unclear to me the reasoning behind running XBeach on the scenario cases for 5, 10, 20 years? As you've just done a linear retreat of the shoreline/profile and there is no account for changes in storminess or SLR [L482-485], are the results not just XBeach present day + retreat (Where a retreat is limited by hard structures such as seawalls)? I was a bit confused on how you did the retreat as well for the cases where structures were present. My general understanding is that a linear retreat (at all elevations) was done which essentially translated the profile intact. If the profile reached a structure, the landward translation stopped at that elevation, but the rest of the profile was allowed to continue to retreat? Or no? Figure 5 suggests that is not the case but it's not clear what was done? In reality, I think if it ran into a structure (like a seawall) the lower elevations would erode more than the linear trend as there would not be the sand from the land to compensate.

Data independence: I have several questions around data independence that I'd like to see addressed. First, while the data set is 60 years long, there are 179 independent storms (43 of these are multi-peak storms). It's not clear to me (from an erosion sense) why you'd split these 43 up into multiple storms to augment your data set to 237 storms (Which is still quite small in terms of BNs). Similarly, on L 155-160 it's again described about the multi-peak storms where a single multi-peak storm is run and the outputs from the cumulative are saved, but also those of the 'first peak' (but the cumulative output after each peak is saved?). Should (ii) not be the peak of each 'sub-peak' in a multi-peak storm and should the output not be the volume (for example) between the 2 peaks, rather than the cumulative over the full event? As an aside - Your wave height cutoffs (98 and 99.5%) are also quite high, so you could lower these and get more smaller storms (say the 95% level – see Masselink et al).

Second, my understanding is that inputs to the BNs are meant to be independent, so closely spaced receptors which are highly correlated shouldn't be included. I couldn't find details on the spacing of the receptors, but they don't look spatially independent to me (Eg. Fig 3). Beuzen et al. (2019 – JGR) I think discussed this and found

the alongshore spacing allowed where correlations dropped off (This would be site specific but in his case it was ∼500m I think). So I suspect you've padded your BN with a bunch of data that's highly correlated which isn't best practice. Similarly, it's not best practice (And I think even discouraged) to augment your data by multiplying your synthetic cases by the number of storms that were in that bin (L144-146). I know it has been done in the past by others (including myself and I've learned from others this was incorrect) but that doesn't make it correct now. I appreciate you are wanting to keep the original distributions but I'm not sure there is a proper way to do this beyond running each case.

How probabilistic is your output? Your BNs (Fig 6 and 7) are quite complex and in some cases, highly discretised. This immensely increases the number of data points needed to ensure the priors are well represented. As the challenge is with much geophysical data, you look to have a lot of near empty bins in your outputs. How many of the relationships are really deterministic rather than probabilistic?

What's the difference between distance to public domain (Fig 7) and distance to beach (Fig 10-12)? I feel they must be similar if not the same so why not use the same classification and binning for the 2?

Specific Comments:

[L74]: 'were' should be 'where' in: "study area were"

[L204]: "Risk to life was also been" should be either 'Risk to live was also' or "Risk to life has also been"

Fig 5 - can you tell the reader what section these are in and the erosion rate used?

[L355] "affecter" should be 'affected'?

[L341] "front a of a" should be "front of a"

[L427] "relation" should be "relationship

---

## Author Comment (AC1) · 29 Oct 2020

I have reviewed the manuscript entitles 'Probabilistic characterisation of coastal storm-induced risks using Bayesian Networks' by Sanuy and Jimenez. Overall the article is of high quality and provides an alternative method for using BN in risk assessment that although it is based on the source-pathway-receptor consequences concept it has some novel methods related with the storm selection.

I believe that the article is of high interest for the journal and well within the journals scope. However, I believe that in order for the manuscript to be accepted some changes need to be addressed more for clarifying some aspects of the work and for providing further information and limitations of the method.

We thank the reviewer for constructive comments. We have performed a thorough revision to address all the comments and incorporated all the suggestions in the manuscript, as detailed below.

**General comments:**

The Abstract of the article although correct is rather general and it is not highlighting the results and the novelty of the work. I believe some addition of more specific results that are present in the discussion will benefit the current version of the abstract.

**[R1]** The Abstract will be modified to incorporate reviewer's suggestions.

Most of my comments are concentrated in the methodology sections this is partly because the method is rather complex and the proposed novelty although important it is not obvious from the begining. The results and discussion sections are very well written and explained with high quality figures that although sometime complex they concentrate a large amount of information.

I have made specific comments in the text where I have questions or doubts my main concerns at the moment is that novelty of the method is not properly described in risk terms. I believe that the BN approach proposed is valid for characterizing the risk for the entire storm climate and not for specific storms as proposed by previous works. However, if this is true it needs to be highlighted by the authors in the abstract and in the title is necessary.

**[R2]** We agree with the reviewer and, in fact, this is one of the main novelties of the work. Following reviewer's suggestion, this will be highlighted in different parts of the text (abstract, introduction, and discussion sections).
We will also propose a modified title: *Characterizing coastal erosion and inundation risks for the entire storm climate using a Bayesian Network*

My secondly concern is related with the scenarios proposed. Some more explanation is needed on why the shoreline retreat is extended to the entire shoreface.

**[R3]** Future (morphological) scenarios have been defined to consider the background evolution of the area. This is important when assessing risks in dynamic areas because if not, the assessment will strictly be valid just for current conditions (small time scale, few years) and, in consequence, of limited validity for coastal (risk) management. This is the key message, the need of updating beach coastal morphology for an effective risk assessment. We will reinforce this message in the text. With respect to how to do it, it will depend on the specific conditions of the area and on the used tool to mimic/simulate such evolution. Whereas there are many different options, we have chosen a simple one by extending shoreline rates of change to reproduce nearshore bathymetric changes, although as mentioned in the work, it can be substituted by a different choice (e.g. by using a morphodynamic model valid at the appropriate time-scale, e.g. Hanson et al. 2003).

In the study area, observed shoreline retreat is the result of the deltaic front reshaping due to a decrease in river sediment supply whereas the wave-induced littoral dynamics maintained its intensity. Transferring this shoreline retreat to the entire active shoreface implies to apply a hypothesis about the shape of long-term (decadal) profile changes. Thus, the most widely used hypothesis used to convert longshore transport – induced shoreline changes to sediment volume is the one applied in one-line models, where a horizontal displacement of the profile from the emerged beach to the closure depth is assumed (e.g. Hanson, 1989). On the contrary, other works on deltaic reduction processes assume that whereas the shoreline is rapidly eroded, the submerged front retreats at a slower rate (e.g. Refaat and Tsuchiya, 1991). This pattern would be consistent with a wedged-shaped change over the closure depth (instead of a parallel one as before). Other type of approach is the one adopted by Stive and de Vriend (1995) when modelling the long-term shoreface evolution. They proposed a varying type of change through the shoreface, from an upper part experiencing a parallel displacement, to a declining/inclining lower shoreface down to the inner shelf limit. As it can be seen, there are different options to reconstruct beach profiles from a modelled/forecasted shoreline, from which we selected one of the most used (albeit not necessarily the best one).

Regardless of the method used, the most important message is that it is necessary to anticipate future coastal morphology in order to make a reliable risk assessment valid not only for current but also for future conditions. We will highlight this in the discussion section and will also introduce a text discussing how the scenarios were constructed (similar to the previous one, but shorter).

References:
Hanson, H.: GENESIS: a generalized shoreline change numerical model, J. Coast. Res., 1-27, 1989.
Hanson, H., Aarninkhof, S., Capobianco, M., Jiménez, J.A., Larson, M., Nicholls, R.J., Plant, N.G., Southgate, H.N., Steetzel, H.J., Stive, M.J.F, and de Vriend, H.J.: Modelling of coastal evolution on yearly to decadal time scales, J. Coast. Res., 19, 4, 790-811, 2003.
Refaat, H., and Tsuchiya, Y.: Formation and reduction processes of river deltas; theory and experiments, Bull. Disaster Prevention Res. Inst. Kyoto Univ., 41, 177-224, 1991.
Stive, M.J.F., and De Vriend, H. J.: Modelling shoreface profile evolution, Mar. Geol., 126(1-4), 235-248, 1995.

**Specific comments:**

LINE 33: source terms are booth the storms and the storm induced hazards.

[R4] Adopting the S-P-R-C framework to analyse the risk induced by erosion/inundation (storm-induced hazards), the source (S) term is just defined by the storms. The pathways (P) of flooding/erosion are composed by the beach, defences and even, in some cases, the coastal floodplain. In fact, pathway and receptor (R) can be considered as relative definitions since they may simultaneously function as pathways to "landward" receptors and as receptors in their own right (e.g. Narayan et al. 2012). We shall slightly rephrase this paragraph in the text for clarification.

LINES 53-58: Plomaritis et al 2018 select the events using the same methods as Poehekke et al., 2016. The method is based on a series of copula applications using Hs as a main parameter. I don0t think that this method can be consider non-probabilistic but indeed the method can differ. Please explain with more detail the differences in the storm selection. Poehekke et al., 2016 also follows the ideas of response approach with the use of copulas but with triangular storms. I believe that the discussion over the different approaches that the authors provide is very interesting and I would suggest extending it or order for the reader to be better informed on the sometime small but important details.

[R5] Following the reviewer's suggestions, we describe/analyse further the differences between approaches. The reviewer is right in stating that the use of the term "non-probabilistic" to classify the method followed by Poehekke et al'16 and Plomaritis et al'18 is not entirely correct and confusing. We have modified the text to avoid such confusion.

The above methods use copulas to statistically represent storms, which are the events (drivers) that induce the analysed hazards. Adopting a strict response-approach involves calculating the induced hazards for the entire storm climate and performing the statistical analysis on the results obtained in terms of hazards/impacts. This difference is especially relevant when analysed hazards depend on multiple storm variables which are not necessarily correlated and not included in their definition through copulas. Moreover, the mentioned works use a selected group of events, instead of a set representing the storm climate.

The reference Duo et al., needs updating.

[R6] Duo, E., Sanuy, M., Jiménez, JA, Ciavola, P. 2020. How Good Are Symmetric Triangular Synthetic Storms to Represent Real Events for Coastal Hazard Modelling. *Coastal Engineering*, 159, 103728.

Study area: Provide the names of the areas in Figure 1 not only the code. Now they is given in Discussion but the codes are used before. I think some information of the areas and the logic behind the separation could be interesting.

[R7] We prefer to do not include names in Figure 1 so as not to "overload" it. However, we have included a text in the study area section in which we give the full name of each sector and give the reasons for their selection (this text was included in section 3.4 in the original version of the manuscript).

LINE 95: I think the paper Sanuy et al. (2018) is not in the reference list.

[R8] Sanuy, M., Duo, E., Wiebke, Jäger, W, Ciavola, P., Jiménez, JA. (2018) Linking source with consequences of coastal storm impacts for climate change and risk reduction scenarios for Mediterranean sandy beaches. *NHESS*, 18, 1825-1847.

LINE 143: Provide the number or persetnage of empty groups

**[R9]** This will be provided, see also **[R10]**

LINES 174-175: How many storms per bin you have in the subset group and which are the output paramters you test? My understanding so far is that you have one storm per group in the subset so, I am not sure how you calculate the variance per bin. Are you evaluate the BN output or input with the equations 1 and 2 or the entire BN?

**[R10]** This question is related with the previous comment. The subset method fills with one storm all combinations showed in Table 1 that have at least one historical event. Some combinations remain empty and this will now be introduced following [R9]. Then, the subset is used to fill the BN, which, as shown in Figure 6, has a different number of bins per variable than classes depicted in Table 1, leading to more than one event in many variable combinations.

The variance per bin is calculated following Bityukov et al., 2013, where the observed standard deviation per bin is estimated with the observed value per bin (i.e., $n_{ik} = \sigma_{ik}$ in eq. 1).

We evaluate both BN input and output variables with equations 1 and 2 (now they can be interpreted from Table 5 and Results Figures). We perform the evaluation on (i) unconstrained output, (ii) output constrained to given input combinations and (iii) input constrained to a given output. In the modified version of the manuscript, the evaluated variables will be detailed, and Table 5 will be adapted to help the correct interpretation of the method.

Hazard Assessment: Which are the indicator (model output parameter) you use for each hazard **[R11]** The XBeach model outputs used are *maxzs* for water depth (inundation hazard) and *sedero* for erosion. They will be mentioned in the revised version of the manuscript.

LINES 194-198: The area characteristics can be put in the study area. See my previous comment. **[R12]** Done. See also [R7].

LINES 246-248: Given the steep slopes of the study area I understand the extrapolation of the shoreline retreat values to the upper beach (-2 to -4 m) but continues retreat up to -8 suggest a huge amount of sediment loss and that all sediment from the upper beach is removed by longshore drift. I am not an expert on Catalan coast but some additional justification for the selected scenarios must be provided.

**[R13]** When building the morphological scenarios, we are using recorded decadal-scale shoreline rates of displacement, that for the study area are mostly controlled by longshore sediment transport (e.g. Jiménez et al. 2018). The objective of the extrapolation was to build "possible coastal morphologies" to illustrate future changes in coastal risk associated with morphodynamic changes. We adopted this simple approach in absence of a robust criteria to select a different one. This point has been extensively covered above in [R3] and, as mentioned there, we will include this point in the discussion section to let readers to make their own choice when applying the method to a given case.

LINE 272: Why the storm parameters are linked in Figure 6? How is te term of previous energy is incorporated in the BN?
**[R14]** The storm parameters are linked so that empty combinations of source characteristics do not propagate noise into the outputs. The term "previous energy" will be removed from the BN (figure and description) as it is not used in the present study.

LINES 274-277: The central variables i and ii are not shown in Figure 6. Please provide more details. Explane where the estimation of the total number of receptors is done, in the BN or before?

**[R15]** In the revised version, Figure 6 will be adapted to show the two variables (and also to remove "previous energy" as mentioned above). The estimation is done before, crossing XBeach output with receptor polygon data, and introduced as an additional variable, at each receptor, that captures the overall number of affected receptors per storm peak. It allows for the assessment, in the same network, of the relation between source characteristics and extension of the impacts, although the presented results put the focus on other variable dependencies found more relevant. A phrase will be introduced in this part to clarify.

LINES 420-421: What are the advantages of this fully probabilistic BN? I suppose that the previous papers were focused on the individual storm assessment while here is attempted an integrated assessment of the storm conditions. If this is correct it has to be stated and event introduced in the abstract.

**[R16]** This has been raised by the reviewer in previous comments. We have introduced some changes in the text (abstract, introduction, discussion) to explicitly mention that the representation of the entire wave climate, to obtain integrated or conditioned risk-oriented results, is the advantage of the presented BN.

---

## Author Comment (AC2) · 29 Oct 2020

I have reviewed the manuscript entitles 'Probabilistic characterization of coastal storm induced risks using Bayesian Networks' by Sanuy and Jimenez. Overall the article is very well written and of high quality. It presents a new framework/ approach using the SPRC framework to examine coastal vulnerability to erosion and inundation at an area within the Spanish coastline exposed to Mediterranean storms. The methodology uses Bayesian Networks to take the SPRC inputs/outputs to create a probabilistic outcome of risk assessment. I believe that the article is well within the journals scope and will be of interest to the readers. However, I believe some changes are needed and points clarified as detailed below.

We thank the reviewer for constructive comments. We have performed a thorough revision to address all the comments and incorporated all the suggestions in the manuscript, as detailed below.

General comments:

Unclear to me the reasoning behind running XBeach on the scenario cases for 5, 10, 20 years? As you've just done a linear retreat of the shoreline/ profile and there is no account for changes in storminess or SLR [L482-485], are the results not just XBeach present day + retreat (Where a retreat is limited by hard structures such as seawalls)? I was a bit confused on how you did the retreat as well for the cases where structures were present. My general understanding is that a linear retreat (at all elevations) was done which essentially translated the profile intact. If the profile reached a structure, the landward translation stopped at that elevation, but the rest of the profile was allowed to continue to retreat? Or no? Figure 5 suggests that is not the case but it's not clear what was done? In reality, I think if it ran into a structure (like a seawall) the lower elevations would erode more than the linear trend as there would not be the sand from the land to compensate.

[R1] XBeach was run for different scenarios (5, 10, 20 y) to assess how expected changes in geomorphology may affect future risks. This may be relevant for decadal-scale retreating areas where (a given) current morphology is only representative of a relatively short (few years) period. We did not include changes in storminess since for the study area (NW Mediterranean) existing projections do not predict significant changes in storminess. We will include a paragraph where this is explicitly stated. Moreover, we will also recommend to perform the analysis using corresponding future storm climates when existing projections indicate a significant change in storminess.

These simulations are not exactly equal to "present day scenario" + "retreat" since the study site has not a homogeneous alongshore behaviour. Thus, the area has been divided (in terms of its decadal scale behaviour) in three different sectors, each one with its corresponding (and different) retreat rate. As a result of this, the alongshore configuration of the delta is slightly different across scenarios, with differences increasing with time due to the cumulative contribution of the background evolution. This change in morphology may affect alongshore

processes and therefore the coastal response to storms (which is resolved with the 2DH - XBeach model).

With respect to the situation when the profile reaches a fixed structure limiting the landward translation, we have assumed that, locally, the beach has disappeared and the profile does not continue to retreat. We recognize that beach behavior in front of seawalls/revetments is more complicated than this, with different processes taking place at different time scales which may affect beach profiles in front of exposed seawalls (Kraus, 1988). In fact, the observation raised by the reviewer on a larger erosion of the lower elevations due to a lack of compensation of material from the emerged part of the beach is one of the typical ones when cross-shore processes are being considered (e.g. Dean, 1986). In spite of this, existing works have documented different responses under different situations. Thus, whereas variations in hydrodynamics and sediment transport at short-term scale have been reported in front of exposed revetments (e.g. Miles et al. 2001), other authors have found that, in spite of differences in short-term behavior, long-term volume erosion rates are not higher in front of seawalls (e.g. Basco et al. 1997).

References:
Basco, D. R., Bellomo, D. A., Hazelton, J. M., & Jones, B. N. (1997). The influence of seawalls on subaerial beach volumes with receding shorelines. *Coastal Engineering*, *30*(3-4), 203-233.
Dean, R. G. (1986). Coastal armoring: effects, principles and mitigation. In: *Proc 20th ICCE*, ASCE, 1843-1857.
Kraus, N. C. (1988). The effects of seawalls on the beach: an extended literature review. *Journal of Coastal Research*, SI4, 1-28.
Miles, J. R., Russell, P. E., & Huntley, D. A. (2001). Field measurements of sediment dynamics in front of a seawall. *Journal of Coastal Research*, 195-206.

Thus, the answer given to reviewer 1 on assumptions to simulate the profile retreat (R3) is also valid for this comment, and we replicate here:

Future (morphological) scenarios have been defined to consider the background evolution of the area. This is important when assessing risks in dynamic areas because if not, the assessment will strictly be valid just for current conditions (small time scale, few years) and, in consequence, of limited validity for coastal (risk) management. This is the key message, the need of updating beach coastal morphology for an effective risk assessment. We will reinforce this message in the text. With respect to how to do it, it will depend on the specific conditions of the area and on the used tool to mimic/simulate such evolution. Whereas there are many different options, we have chosen a simple one by extending shoreline rates of change to reproduce nearshore bathymetric changes, although as mentioned in the work, it can be substituted by a different choice (e.g. by using a morphodynamic model valid at the appropriate time-scale, e.g. Hanson et al. 2003).

In the study area, observed shoreline retreat is the result of the deltaic front reshaping due to a decrease in river sediment supply whereas the wave-induced littoral dynamics maintained its intensity. Transferring this shoreline retreat to the entire active shoreface implies to apply a hypothesis about the shape of long-term (decadal) profile changes. Thus, the most widely used hypothesis used to convert longshore transport – induced shoreline changes to sediment volume is the one applied in one-line models, where a horizontal displacement of the profile from the emerged beach to the closure depth is assumed (e.g. Hanson, 1989). On the contrary, other works on deltaic reduction processes assume that whereas the shoreline is rapidly eroded,

the submerged front retreats at a slower rate (e.g. Refaat and Tsuchiya, 1991). This pattern would be consistent with a wedged-shaped change over the closure depth (instead of a parallel one as before). Other type of approach is the one adopted by Stive and de Vriend (1995) when modelling the long-term shoreface evolution. They proposed a varying type of change through the shoreface, from an upper part experiencing a parallel displacement, to a declining/inclining lower shoreface down to the inner shelf limit. As it can be seen, there are different options to reconstruct beach profiles from a modelled/forecasted shoreline, from which we selected one of the most used (albeit not necessarily the best one).

Regardless of the method used, the most important message is that it is necessary to anticipate future coastal morphology in order to make a reliable risk assessment valid not only for current but also for future conditions. We will highlight this in the discussion section and will also introduce a text discussing how the scenarios were constructed (similar to the previous one, but shorter).

References:
Hanson, H.: GENESIS: a generalized shoreline change numerical model, J. Coast. Res., 1-27, 1989.
Hanson, H., Aarninkhof, S., Capobianco, M., Jiménez, J.A., Larson, M., Nicholls, R.J., Plant, N.G., Southgate, H.N., Steetzel, H.J., Stive, M.J.F, and de Vriend, H.J.: Modelling of coastal evolution on yearly to decadal time scales, J. Coast. Res., 19, 4, 790-811, 2003.
Refaat, H., and Tsuchiya, Y.: Formation and reduction processes of river deltas; theory and experiments, Bull. Disaster Prevention Res. Inst. Kyoto Univ., 41, 177-224, 1991.
Stive, M.J.F., and De Vriend, H. J.: Modelling shoreface profile evolution, Mar. Geol., 126(1-4), 235-248, 1995.

Data independence: I have several questions around data independence that I'd like to see addressed.

First, while the data set is 60 years long, there are 179 independent storms (43 of these are multi-peak storms). It's not clear to me (from an erosion sense) why you'd split these 43 up into multiple storms to augment your data set to 237 storms (Which is still quite small in terms of BNs). Similarly, on L 155-160 it's again described about the multi-peak storms where a single multi-peak storm is run and the outputs from the cumulative are saved, but also those of the 'first peak' (but the cumulative output after each peak is saved?). Should (ii) not be the peak of each 'sub-peak' in a multi-peak storm and should the output not be the volume (for example) between the 2 peaks, rather than the cumulative over the full event? As an aside - Your wave height cutoffs (98 and 99.5%) are also quite high, so you could lower these and get more smaller storms (say the 95% level – see Masselink et al).

[R2] *With respect to creating a dataset based on storm peaks instead of storms.*
Individual storm events have been identified and isolated by using the P.O.T method that ensures they are independent. Then, from there, any storm consisting in more than one peak is treated by its individual (cumulative) peaks, as the idea was to create a dataset of storm peaks (not to artificially augment the dataset with additional storms). For each peak, we retain its duration, together with the total accumulated event duration, and the previous energy (i.e. single-peak storms are always characterised as peaks with "peak duration" equal to "event duration" and with "zero previous energy"). This was done for a parallel analysis on morphodynamic response where we found that peak sequencing was a key aspect to predict local beach retreats. These variables were included in the network to assess their impact into output risk variables, but for the sake of simplicity only a selection of them, focusing on other variables, is presented here, and due to this they have been shortly described, which could

generate some confusion. We will give more extension to variable description in the revised version.

The reviewer is fully right affirming that each "sub-peak" should be considered (not only the first). In fact, the original dataset contains ALL sub-peaks. Text in L155-160 refers to the fact that in order to create the subsets for the future scenarios, and with the objective of reducing the number of time-consuming simulations, the first peak of a multipeak storm is also used as a proxy of "single-peak-storms of the same characteristics". We will rephrase part of the "Storm characterisation" section to clarify this point.

*With respect to threshold selection.*
The used thresholds when applying the P.O.T method (98% and 99.5% percentiles of the wave height distribution) have been previously used in other works in the study area (Sanuy et al., 2019; Sanuy and Jiménez 2020). Obtained results (identified storms) have been compared with storm conditions associated with representative storm classes (Mendoza et al., 2011) and they fit with values obtained therein for Class 1 and Class 3 storms. Class 1 storms have the minimum Hs historically used in the Mediterranean as threshold for extreme events (2 m), while Class 3 events have the minimum Hs that actually induces hazardous coastal response. This is equivalent to define storms as starting and ending with a Class 1 magnitude, and having at least Class 3 at the peak. This permits to assure that all included events will induce a relevant coastal response from the risk-oriented standpoint.
The obtained event density of 3.5 events/year is appropriate for extreme-climate analysis, and lowering the threshold would increase this frequency by including not too extreme events which would not significantly contribute to overall risk. Due to this, we will maintain the proposed thresholds which have been locally validated for this use. In spite of this, we will stress the meaning of the thresholds, specifying that the levels are site-dependent both in the Storm Characterization" and "Discussion" sections.

References:
Mendoza, E. T., Jimenez, J. A. and Mateo, J. 2011. A coastal storms intensity scale for the Catalan sea (NW Mediterranean), *Nat . Hazards Earth Syst . Sci*, 11, 2453–2462.
Sanuy, M., Jiménez, J. A., Ortego, M. I. and Toimil, A. 2019: Differences in assigning probabilities to coastal inundation hazard estimators: Event versus response approaches, *J. Flood Risk Manag*., 13, e12557.
Sanuy, M., Jiménez, J. A. and Plant, N. 2020. A Bayesian Network methodology for coastal hazard assessments on a regional scale: The BN-CRAF, *Coast. Eng*., 1572019, 1–10.

Second, my understanding is that inputs to the BNs are meant to be independent, so closely spaced receptors which are highly correlated shouldn't be included. I couldn't find details on the spacing of the receptors, but they don't look spatially independent to me (Eg. Fig 3). Beuzen et al. (2019 – JGR) I think discussed this and found the alongshore spacing allowed where correlations dropped off (This would be site specific but in his case it was _500m I think). So I suspect you've padded your BN with a bunch of data that's highly correlated which isn't best practice.

[R3] This answer is related with R5 (see below). Beuzen et al. (2019-JGR) deals with morphological patterns at regional scale (~400 km). They aim for a predictive BN and therefore they cannot allow for correlations in the input. Indeed, distances would be case specific, and places as the Tordera Delta (curvilinear shoreline with significant alongshore morphological variability, and beach-structure interactions inducing local processes such as flanking effects) these distances would be much lower, as we found by analysing the morphological response

sector by sector (analysed in a companion morphodynamic oriented-paper, currently under review).

However, this is out of the scope of the current paper, which is risk-oriented. Here, the individual receptors must be represented as they indicate the spatial extension and magnitude of the impacts induced by a given coastal response (e.g. its not the same from the risk perspective 100 m of eroded dune in front of 1 receptor than the same 100 m of eroded dune in front of 2 lines of 20 receptors). Thus, we have adopted the Source-Pathway-Receptor-Consequence (SPRC) scheme as in Poelhekke et al 2016, Jäger et al., 2018, Plomaritis et al., 2018 and Sanuy et al., 2018 to account for the actual receptor density and typology at the local scale.

References:
Jäger, W. S., Christie, E. K., Hanea, A. M., den Heijer, C. and Spencer, T. 2018: A Bayesian network approach for coastal risk analysis and decision making, Coast. Eng., 134, 48-61.
Plomaritis, T. A., Costas, S. and Ferreira, Ó. 2018: Use of a Bayesian Network for coastal hazards, impact and disaster risk reduction assessment at a coastal barrier (Ria Formosa, Portugal), Coast. Eng., 134, 134-147.
Poelhekke, L., Jäger, W. S., van Dongeren, A., Plomaritis, T. A., McCall, R. and Ferreira, Ó.: 2016. Predicting coastal hazards for sandy coasts with a Bayesian Network, Coast. Eng., 118, 21–34.
Sanuy, M., Duo, E., Jäger, W. S., Ciavola, P., and Jiménez, J. A. 2018: Linking source with consequences of coastal storm impacts for climate change and risk reduction scenarios for Mediterranean sandy beaches, Nat. Hazards Earth Syst. Sci., 18, 1825–1847.

Similarly, it's not best practice (And I think even discouraged) to augment your data by multiplying your synthetic cases by the number of storms that were in that bin (L144-146). I know it has been done in the past by others (including myself and I've learned from others this was incorrect) but that doesn't make it correct now. I appreciate you are wanting to keep the original distributions but I'm not sure there is a proper way to do this beyond running each case.

[R4] We agree that this is a shortcoming compared to running all cases. However, this method was proposed to reduce computational time when generating future scenarios (which are affected to other additional uncertainties as well). In statistical terms, the method behaves consistently and it is validated by comparing the distributions obtained with the subset with those of the original dataset (for the baseline scenario). This means that for the purposes presented in the current work, i.e. obtaining risk-oriented variable distributions, the obtained subsets can be considered statistically similar to the original dataset (although for more detailed analyses, such as morphological-oriented ones, this may not be enough).

How probabilistic is your output? Your BNs (Fig 6 and 7) are quite complex and in some cases, highly discretised. This immensely increases the number of data points needed to ensure the priors are well represented. As the challenge is with much geophysical data, you look to have a lot of near empty bins in your outputs. How many of the relationships are really deterministic rather than probabilistic?

[R5] We understand our BN is probabilistic in the sense that it is is used to adopt the SPRC model by using a probabilistic representation of the source (i.e. a probabilistic representation of the storm climate of the study site).
The reviewer is right when pointing out the complexity of the BN, and the data requirements that this involves to properly fill it. In this case, all Source-related parent variables are connected between them (differently e.g. to Beuzen et al., 2019) to ensure that when conditioning is made on these variables all other priors are updated so as not to have noise propagation onto the output variables. In this sense, our BN would fit into the descriptive BN category according to

Beuzen et al. (2018). This does not mean that the output is not probabilistic (which is by the schematization of the SPRC and the treatment of the Source) but that the main purpose of the BN will not be a predictive one, as e.g. in Beuzen et al. (2019).

We shall properly address this point in the Discussion section of the reviewed version of the manuscript. Additionally, the main (and novelty with respect to previous works) purpose of the BN, which is the probabilistic representation of the source, will be also better stressed, as suggested also by reviewer 1 (see answers to reviewer 1 general comments).

References:
Beuzen, T., Splinter, K.D., Marshall, L.A., Turner, I.L., Harley, M.D., Palmsten, M.L. 2018. Bayesian Networks in coastal engineering: Distinguishing descriptive and predictive applications. Coast. Eng. 135, 16–30.

Beuzen, T., Harley, M. D., Splinter, K. D., & Turner, I. L. 2019. Controls of variability in berm and dune storm erosion. *Journal of Geophysical Research: Earth Surface*, *124*(11), 2647-2665.

What's the difference between distance to public domain (Fig 7) and distance to beach (Fig 10-12)? I feel they must be similar if not the same so why not use the same classification and binning for the 2?

[R6] Indeed, they are the same since the line of public domain is running along the inner limit of the beach. The name in figure 7 (and related text) will be changed. The binning is actually the same, but in figures 10-12 the outputs of two lowest bins are summed under the name "Beach to 10m", and the outputs of the four highest bins are summed under the name "> 75 m"

Specific Comments:
[L74]: 'were' should be 'where' in: "study area were"
[R7] This will be addressed in the updated version of the manuscript.

[L204]: "Risk to life was also been" should be either 'Risk to live was also' or "Risk to life has also been"
[R8] This will be addressed in the updated version of the manuscript.

Fig 5 - can you tell the reader what section these are in and the erosion rate used?
[R9] This will be included in the new version of the figure.

[L355] "affecter" should be 'affected'?
[R10] This will be addressed in the updated version of the manuscript.

[L341] "front a of a" should be "front of a"
[R11] This will be addressed in the updated version of the manuscript.

[L427] "relation" should be "relationship
[R12] This will be addressed in the updated version of the manuscript.

---

## Author Response (AR1)

**Combined answer to reviewers with changes in manuscript**

**Anonymous Referee #1**

I have reviewed the manuscript entitles 'Probabilistic characterisation of coastal storm-induced risks using Bayesian Networks' by Sanuy and Jimenez. Overall the article is of high quality and provides an alternative method for using BN in risk assessment that although it is based on the source-pathway-receptor consequences concept it has some novel methods related with the storm selection.

I believe that the article is of high interest for the journal and well within the journals scope. However, I believe that in order for the manuscript to be accepted some changes need to be addressed more for clarifying some aspects of the work and for providing further information and limitations of the method.

We thank the reviewer for the detailed review and constructive comments on the manuscript. We have performed a thorough revision to address all the comments, as detailed below.

**General comments:**
The Abstract of the article although correct is rather general and it is not highlighting the results and the novelty of the work. I believe some addition of more specific results that are present in the discussion will benefit the current version of the abstract.

**[R1.1]** The Abstract has been modified to incorporate reviewer's suggestions.

**The following text has been added:**
**Abstract (L15)** "*As an example, storms with smaller waves and from secondary incoming direction will increase erosion and inundation risks at the study area*"

**Abstract (L18)** "*Under current conditions, high and moderate inundation risks, and direct exposure to erosion can be reduced with a small coastal setback (~10 m), which needs to be increased up to 20-55 m to be efficient under future scenarios (+20 years).*"

Most of my comments are concentrated in the methodology sections this is partly because the method is rather complex and the proposed novelty although important it is not obvious from the begining. The results and discussion sections are very well written and explained with high quality figures that although sometime complex they concentrate a large amount of information.

I have made specific comments in the text where I have questions or doubts my main concerns at the moment is that novelty of the method is not properly described in risk terms. I believe that the BN approach proposed is valid for characterizing the risk for the entire storm climate and not for specific storms as proposed by previous works. However, if this is true it needs to be highlighted by the authors in the abstract and in the title is necessary.

**[R1.2]** We agree with the reviewer and, in fact, this is one of the main novelties of the work. Following reviewer's suggestion, this is highlighted in different parts of the text.

**Abstract (L11):** *One of the main differences of the developed BN framework is that it includes the entire storm climate (all recorded storm events, 179 in the study case) to retrieve the integrated and conditioned risk-oriented results at individually identified receptors (about 4,000 in the study case).*

**Introduction (L54):** **The inclusion in the BN of simulation results from a long dataset of storms allows for a fully stochastic assessment in terms of wave climate characterisation.** *This is a novelty with respect to existing studies (e.g. Van Verseveld et al., 2015; Plomaritis et al., 2018; Ferreira et al. 2019; Sanuy et al., 2018). Although some of these studies introduce copula assessments on source (storm) characteristic variables to generate synthetic events, the training subsets aimed to covering the whole range of possible storm conditions rather than statistically representing the existing storm climate*

**Conclusion (L535):** *This resulted in a full representation of the storm climate (source) leading to probabilistic characterisation of risks that accounted for climate (storms) and geographic (receptor location) related variabilities, as the BN training followed the response approach (i.e. the simulation of the coastal response for all identified storms).*

My secondly concern is related with the scenarios proposed. Some more explanation is needed on why the shoreline retreat is extended to the entire shoreface.

**[R1.3]** Future (morphological) scenarios have been defined to consider the background evolution of the area. This is important when assessing risks in dynamic areas because if not, the assessment will strictly be valid just for current conditions (small time scale, few years) and, in consequence, of limited validity for coastal (risk) management. This is the key message, the need of updating beach coastal morphology for an effective risk assessment. We will reinforce this message in the text. With respect to how to do it, it will depend on the specific conditions of the area and on the used tool to mimic/simulate such evolution. Whereas there are many different options, we have chosen a simple one by extending shoreline rates of change to reproduce nearshore bathymetric changes, although as mentioned in the work, it can be substituted by a different choice (e.g. by using a morphodynamic model valid at the appropriate time-scale, e.g. Hanson et al. 2003).

In the study area, observed shoreline retreat is the result of the deltaic front reshaping due to a decrease in river sediment supply whereas the wave-induced littoral dynamics maintained its intensity. Transferring this shoreline retreat to the entire active shoreface implies to apply a hypothesis about the shape of long-term (decadal) profile changes. Thus, the most widely used hypothesis used to convert longshore transport – induced shoreline changes to sediment volume is the one applied in one-line models, where a horizontal displacement of the profile from the emerged beach to the closure depth is assumed (e.g. Hanson, 1989). On the contrary, other works on deltaic reduction processes assume that whereas the shoreline is rapidly eroded, the submerged front retreats at a slower rate (e.g. Refaat and Tsuchiya, 1991). This pattern would be consistent with a wedged-shaped change over the closure depth (instead of a parallel one as before). Other type of approach is the one adopted by Stive and de Vriend (1995) when modelling the long-term shoreface evolution. They proposed a varying type of change through the shoreface, from an upper part experiencing a parallel displacement, to a declining/inclining lower shoreface down to the inner shelf limit. As it can be seen, there are different options to reconstruct beach profiles from a modelled/forecasted shoreline, from which we selected one of the most used (albeit not necessarily the best one).

Regardless of the method used, the most important message is that it is necessary to anticipate future coastal morphology in order to make a reliable risk assessment valid not only for current but also for future conditions. We have highlighted this in the discussion section and also introduced a text discussing how the scenarios were constructed (similar to the previous one, but shorter).

References:

Hanson, H.: GENESIS: a generalized shoreline change numerical model, J. Coast. Res., 1-27, 1989.
Hanson, H., Aarninkhof, S., Capobianco, M., Jiménez, J.A., Larson, M., Nicholls, R.J., Plant, N.G., Southgate, H.N., Steetzel, H.J., Stive, M.J.F, and de Vriend, H.J.: Modelling of coastal evolution on yearly to decadal time scales, J. Coast. Res., 19, 4, 790-811, 2003.
Refaat, H., and Tsuchiya, Y.: Formation and reduction processes of river deltas; theory and experiments, Bull. Disaster Prevention Res. Inst. Kyoto Univ., 41, 177-224, 1991.
Stive, M.J.F., and De Vriend, H. J.: Modelling shoreface profile evolution, Mar. Geol., 126(1-4), 235-248, 1995.

**Added text in the manuscript:**
**Scenario definition (L250):** *Here, future morphological scenarios are defined to consider the background erosion in the area*

**Scenario definition (L268):** *This hypothesis about the shape of long-term (decadal) profile changes follows the hypothesis applied in shoreline evolution models, i.e. a parallel displacement of the active profile from the emerged beach down to the depth of closure (e.g. Hanson, 1989)*

**Scenario definition (L274):** *When the shoreline reaches a fixed structure limiting the landward translation, it is assumed that, locally, the beach disappears and, in consequence, no further profile retreat will occur.*

**Discussion (L511):** *It has to be mentioned that to build these morphological scenarios, it is necessary to "forecast" future configurations of the shallow water bathymetry. In this work, this was done by extending shoreline displacements down to the depth of closure by assuming a simple parallel displacement of the active inner profile, which is compatible with the usual hypothesis applied in mid-term shoreline models. However, other profile change modes could also be applied, such as a wedged-shaped change over the closure depth to simulate a slower retreat of the delta front in comparison with faster shoreline changes (e.g. Refaat and Tsuchiya, 1991). In both cases, their morphological consequences are limited to the shallowest and faster part of the shoreface and, in consequence, are strictly applicable to expected mid-term (decadal) changes. Building longer-term morphological scenarios would require to consider other options since the depth limiting significant changes in the beach profile will extend further with time scale (e.g. Cowell et al. 1999). In this line, Stive and de Vriend (1995) proposed a long-term shoreface evolution model that considers a varying type of change through the shoreface, from an upper part experiencing a parallel displacement, to a declining/inclining lower shoreface down to the inner shelf limit.*

*In the case of structures/barriers being exposed at the shoreline along the study area due to background erosion, we have assumed that, locally, the active profile will not retreat further once the beach had disappeared. In the event of such situation, the structure would be*

*subjected to the highest possible risk and as so would be classified in the framework. Further bottom variations in front of the structure which may lead to its collapse due to scouring will not modify this classification.*

*In any case, it has to be considered that building future morphological scenarios to forecast the evolution of coastal risks at long-term scales will add uncertainty to the analysis, in addition to that associated with expected varying climatic forcing, since long-term morphodynamic modelling integrating all relevant processes is still an unsolved issue (e.g. Ranasinghe, 2020).*

**Specific comments:**
LINE 33: source terms are booth the storms and the storm induced hazards.
**[R1.4]** Adopting the S-P-R-C framework to analyse the risk induced by erosion/inundation (storm-induced hazards), the source (S) term is just defined by the storms. The pathways (P) of flooding/erosion are composed by the beach, defences and even, in some cases, the coastal floodplain. In fact, pathway and receptor (R) can be considered as relative definitions since they may simultaneously function as pathways to "landward" receptors and as receptors in their own right (e.g. Narayan et al. 2012). We slightly rephrased this paragraph in the text for clarification.

**Modified paragraph:**
**Introduction (L35):** *When applied to storm-induced coastal risks, it is generally schematised in terms of a source (storms), that propagates and interacts with a pathway (beach or coastal morphology) where hazards (i.e. inundation and erosion) are generated. These affect the receptors (elements of interest), inducing different consequences.*

LINES 53-58: Plomaritis et al 2018 select the events using the same methods as Poehekke et al., 2016. The method is based on a series of copula applications using Hs as a main parameter. I don0t think that this method can be consider non-probabilistic but indeed the method can differ. Please explain with more detail the differences in the storm selection. Poehekke et al., 2016 also follows the ideas of response approach with the use of copulas but with triangular storms. I believe that the discussion over the different approaches that the authors provide is very interesting and I would suggest extending it or order for the reader to be better informed on the sometime small but important details.

**[R1.5]** Following the reviewer's suggestions, we describe/analyse further the differences between approaches. The reviewer is right in stating that the use of the term "non-probabilistic" to classify the method followed by Poehekke et al'16 and Plomaritis et al'18 is not entirely correct and confusing. We have modified the text to avoid such confusion.

The above methods use copulas to statistically represent storms, which are the events (drivers) that induce the analysed hazards. Adopting a strict response-approach involves calculating the induced hazards for the entire storm climate and performing the statistical analysis on the results obtained in terms of hazards/impacts. This difference is especially relevant when analysed hazards depend on multiple storm variables which are not necessarily correlated and not included in their definition through copulas. Moreover, the mentioned works use a selected group of events, instead of a set representing the storm climate.

**Modified paragraph:**
**Introduction (L56):** *Although some of these studies introduce copula assessments on source (storm) characteristic variables to generate synthetic events, the training subsets aimed to*

*covering the whole range of possible storm characteristic rather than statistically representing the existing storm climate.*

The reference Duo et al., needs updating.
[R1.6] Update in the reference list:
**Duo, E., Sanuy, M., Jiménez, JA, Ciavola, P. 2020. How Good Are Symmetric Triangular Synthetic Storms to Represent Real Events for Coastal Hazard Modelling. *Coastal Engineering*, 159, 103728.**

Study area: Provide the names of the areas in Figure 1 not only the code. Now they is given in Discussion but the codes are used before. I think some information of the areas and the logic behind the separation could be interesting.
[R1.7] We prefer to do not include names in Figure 1 so as not to "overload" it. However, we have included a text in the "Study area" section in which we provide the full name of each sector and reasons for their selection (this text was included in section 3.4 in the original version of the manuscript).

**Modification in the manuscript:**
**The paragraph describing the sectors that was previously located in the Risks (Methods) section has been moved without changes to the Study Area section.**

LINE 95: I think the paper Sanuy et al. (2018) is not in the reference list.
[R1.8] Added to the reference list:
**Sanuy, M., Duo, E., Wiebke, Jäger, W, Ciavola, P., Jiménez, JA. (2018) Linking source with consequences of coastal storm impacts for climate change and risk reduction scenarios for Mediterranean sandy beaches. *NHESS*, 18, 1825-1847.**

LINE 143: Provide the number or persetnage of empty groups
[R1.9] Done, see also [R1.10]

**Storm characterisation (L157): *Each storm from the dataset falls into one of the resulting 5 × 4 × 3 × 3 = 188 combinations of bulk characteristics. Some combinations are populated with storms (48), while others are empty groups (140), i.e. storm characteristics that have not been recorded and, therefore, not present in the storm dataset.***

LINES 174-175: How many storms per bin you have in the subset group and which are the output paramters you test? My understanding so far is that you have one storm per group in the subset so, I am not sure how you calculate the variance per bin. Are you evaluate the BN output or input with the equations 1 and 2 or the entire BN?

[R1.10] This question is related with the previous comment. The subset method fills with one storm all combinations showed in Table 1 that have at least one historical event. Some combinations remain empty and this will now be introduced following [R1.9]. Then, the subset is used to fill the BN, which, as shown in Figure 6, has a different number of bins per variable than classes depicted in Table 1, leading to more than one event in many variable combinations.

The variance per bin is calculated following Bityukov et al., 2013, where the observed standard deviation per bin is estimated with the observed value per bin (i.e., $n_{ik} = \sigma_{ik}$ in eq. 1).

We evaluate both BN input and output variables with equations 1 and 2 (now they can be interpreted from Table 5 and Results Figures). We perform the evaluation on (i) unconstrained output, (ii) output constrained to given input combinations and (iii) input constrained to a given output. In the modified version of the manuscript, the evaluated variables are detailed, and Table 5 has been adapted to help the correct interpretation of the method.

**Modifications in the manuscript**
**Storm characterisation (L159):** *This subdivision is only used for the purpose of deriving the subset, allowing finer detail in the source characteristics of the single-peak and multi-peak storms to be selected. Later, the BN will present a coarser binning of such variables, ensuring a better filling of the source variable combinations in the network.*

**Storm characterisation (L194):** *The statistics will be calculated for both BN inputs and outputs (see following sections): (i) the distribution of un-constrained output risk variables; (ii) the distribution of Hs, Tp, duration, direction and water level constrained to the different risk levels per sector; and (iii) the risk distributions per area and conditioned to the distance to inner beach limit. This involves the comparison of more than one variable output (e.g. impact results are always three variables), and therefore, results are given as a mean and standard deviation.*

**Table 5 has been modified naming the variables in it.**

Hazard Assessment: Which are the indicator (model output parameter) you use for each hazard **[R1.11]** The XBeach model outputs used are *maxzs* for water depth (inundation hazard) and *sedero* for erosion. This is mentioned in the revised version of the manuscript.

**Hazards assessment (L209):** *The XBeach model outputs used for the subsequent risk calculations were maxzs for water depth with accompanying u,v components of the water velocity (inundation hazard) and sedero for bed level change (erosion hazard).*

LINES 194-198: The area characteristics can be put in the study area. See my previous comment. **[R1.12]** Done. See also **[R1.7].**

LINES 246-248: Given the steep slopes of the study area I understand the extrapolation of the shoreline retreat values to the upper beach (-2 to -4 m) but continues retreat up to -8 suggest a huge amount of sediment loss and that all sediment from the upper beach is removed by longshore drift. I am not an expert on Catalan coast but some additional justification for the selected scenarios must be provided.

**[R1.13]** When building the morphological scenarios, we are using recorded decadal-scale shoreline rates of displacement, that for the study area are mostly controlled by longshore sediment transport (e.g. Jiménez et al. 2018). The objective of the extrapolation was to build "possible coastal morphologies" to illustrate future changes in coastal risk associated with morphodynamic changes. We adopted this simple approach in absence of a robust criteria to select a different one. This point has been extensively covered above in [R3] and, as mentioned there, we included this point in the discussion section to let readers to make their own choice when applying the method to a given case.

**Modifications in the manuscript: see [R1.3]**

LINE 272: Why the storm parameters are linked in Figure 6? How is te term of previous energy is incorporated in the BN?

**[R1.14]** The storm parameters are linked so that empty combinations of source characteristics do not propagate noise into the outputs.

The term previous energy is now defined in the following added text:

**Storm characterization (L147):** *For each peak, we retain its duration, together with the total accumulated event duration, and the previous energy: e.g. single-peak storms are always characterised as peaks with "peak duration" equal to "event duration" and with "mul"). Although all this information is retained (Figure 2), only event duration together with wave parameters and water level will be used as BN variable here, for the sake of simplicity in a risk-oriented perspective, while more detailed source description may be necessary in morphological analyses.*

LINES 274-277: The central variables i and ii are not shown in Figure 6. Please provide more details. Explane where the estimation of the total number of receptors is done, in the BN or before?

**[R1.15]** In the revised version, Figure 6 will be adapted to show the two variables. The estimation is done before, crossing XBeach output with receptor polygon data, and introduced as an additional variable, at each receptor, that captures the overall number of affected receptors per storm peak. It allows for the assessment, in the same network, of the relation between source characteristics and extension of the impacts, although the presented results put the focus on other variable dependencies found more relevant. A sentence is introduced for further clarification.

**Bayesian Network integration (L302):** *These are counted outside the BN for each simulated storm peak and introduced in the BN as an additional storm characteristic variable*

LINES 420-421: What are the advantages of this fully probabilistic BN? I suppose that the previous papers were focused on the individual storm assessment while here is attempted an integrated assessment of the storm conditions. If this is correct it has to be stated and event introduced in the abstract.

**[R1.16]** This has been raised by the reviewer in previous comments. We have introduced some changes in the text (abstract, introduction, conclusion) to explicitly mention that the representation of the entire wave climate, to obtain integrated or conditioned risk-oriented results, is the advantage of the presented BN.

**Modification in the manuscript: See [R1.2]**

**Anonymous Referee #2**

I have reviewed the manuscript entitles 'Probabilistic characterization of coastal storm induced risks using Bayesian Networks' by Sanuy and Jimenez. Overall the article is very well written and of high quality. It presents a new framework/ approach using the SPRC framework to examine coastal vulnerability to erosion and inundation at an area within the Spanish coastline exposed to Mediterranean storms. The methodology uses Bayesian Networks to take the SPRC inputs/outputs to create a probabilistic outcome of risk assessment. I believe that the article is well within the journals scope and will be of interest to the readers. However, I believe some changes are needed and points clarified as detailed below.

We thank the reviewer for constructive comments. We have performed a thorough revision to address all the comments and incorporated all the suggestions in the manuscript, as detailed below.

General comments:

Unclear to me the reasoning behind running XBeach on the scenario cases for 5, 10, 20 years? As you've just done a linear retreat of the shoreline/ profile and there is no account for changes in storminess or SLR [L482-485], are the results not just XBeach present day + retreat (Where a retreat is limited by hard structures such as seawalls)? I was a bit confused on how you did the retreat as well for the cases where structures were present. My general understanding is that a linear retreat (at all elevations) was done which essentially translated the profile intact. If the profile reached a structure, the landward translation stopped at that elevation, but the rest of the profile was allowed to continue to retreat? Or no? Figure 5 suggests that is not the case but it's not clear what was done? In reality, I think if it ran into a structure (like a seawall) the lower elevations would erode more than the linear trend as there would not be the sand from the land to compensate.

[R2.1] XBeach was run for different scenarios (5, 10, 20 y) to assess how expected changes in geomorphology may affect future risks. This may be relevant for decadal-scale retreating areas where (a given) current morphology is only representative of a relatively short (few years) period. We did not include changes in storminess since for the study area (NW Mediterranean) existing projections do not predict significant changes in storminess. We will include a paragraph where this is explicitly stated. Moreover, we will also recommend to perform the analysis using corresponding future storm climates when existing projections indicate a significant change in storminess.

These simulations are not exactly equal to "present day scenario" + "retreat" since the study site has not a homogeneous alongshore behaviour. Thus, the area has been divided (in terms of its decadal scale behaviour) in three different sectors, each one with its corresponding (and different) retreat rate. As a result of this, the alongshore configuration of the delta is slightly different across scenarios, with differences increasing with time due to the cumulative contribution of the background evolution. This change in morphology may affect alongshore processes and therefore the coastal response to storms (which is resolved with the 2DH - XBeach model).

With respect to the situation when the profile reaches a fixed structure limiting the landward translation, we have assumed that, locally, the beach has disappeared and the profile does not continue to retreat. We recognize that beach behavior in front of seawalls/revetments is more

complicated than this, with different processes taking place at different time scales which may affect beach profiles in front of exposed seawalls (Kraus, 1988). In fact, the observation raised by the reviewer on a larger erosion of the lower elevations due to a lack of compensation of material from the emerged part of the beach is one of the typical ones when cross-shore processes are being considered (e.g. Dean, 1986). In spite of this, existing works have documented different responses under different situations. Thus, whereas variations in hydrodynamics and sediment transport at short-term scale have been reported in front of exposed revetments (e.g. Miles et al. 2001), other authors have found that, in spite of differences in short-term behavior, long-term volume erosion rates are not higher in front of seawalls (e.g. Basco et al. 1997).

In any case it has to be considered that the objective of the framework is not simulate morphodynamic evolution but to assess the expected risk. In the case of structures/barriers being exposed at the shoreline along the study area due to background erosion, the structure would be subjected to the highest possible risk and as so would be classified in the framework. Further bottom variations in front of the structure which may lead to the collapse of the structure due to scouring will not modify this classification, and their prediction is further beyond the objectives of this work.

Thus, the answer given to reviewer 1 on assumptions to simulate the profile retreat **[R1.3]** is also valid for this comment, and we replicate here:

**[R1.3]** Future (morphological) scenarios have been defined to consider the background evolution of the area. This is important when assessing risks in dynamic areas because if not, the assessment will strictly be valid just for current conditions (small time scale, few years) and, in consequence, of limited validity for coastal (risk) management. This is the key message, the need of updating beach coastal morphology for an effective risk assessment. We will reinforce this message in the text. With respect to how to do it, it will depend on the specific conditions of the area and on the used tool to mimic/simulate such evolution. Whereas there are many different options, we have chosen a simple one by extending shoreline rates of change to reproduce nearshore bathymetric changes, although as mentioned in the work, it can be substituted by a different choice (e.g. by using a morphodynamic model valid at the appropriate time-scale, e.g. Hanson et al. 2003).

In the study area, observed shoreline retreat is the result of the deltaic front reshaping due to a decrease in river sediment supply whereas the wave-induced littoral dynamics maintained its intensity. Transferring this shoreline retreat to the entire active shoreface implies to apply a hypothesis about the shape of long-term (decadal) profile changes. Thus, the most widely used hypothesis used to convert longshore transport – induced shoreline changes to sediment

volume is the one applied in one-line models, where a horizontal displacement of the profile from the emerged beach to the closure depth is assumed (e.g. Hanson, 1989). On the contrary, other works on deltaic reduction processes assume that whereas the shoreline is rapidly eroded, the submerged front retreats at a slower rate (e.g. Refaat and Tsuchiya, 1991). This pattern would be consistent with a wedged-shaped change over the closure depth (instead of a parallel one as before). Other type of approach is the one adopted by Stive and de Vriend (1995) when modelling the long-term shoreface evolution. They proposed a varying type of change through the shoreface, from an upper part experiencing a parallel displacement, to a declining/inclining lower shoreface down to the inner shelf limit. As it can be seen, there are different options to reconstruct beach profiles from a modelled/forecasted shoreline, from which we selected one of the most used (albeit not necessarily the best one).

Regardless of the method used, the most important message is that it is necessary to anticipate future coastal morphology in order to make a reliable risk assessment valid not only for current but also for future conditions. We have highlighted this in the discussion section and also introduced a text discussing how the scenarios were constructed (similar to the previous one, but shorter).

Data independence: I have several questions around data independence that I'd like to see addressed.

First, while the data set is 60 years long, there are 179 independent storms (43 of these are multi-peak storms). It's not clear to me (from an erosion sense) why you'd split these 43 up into multiple storms to augment your data set to 237 storms (Which is still quite small in terms of BNs). Similarly, on L 155-160 it's again described about the multi-peak storms where a single multi-peak storm is run and the outputs from the cumulative are saved, but also those of the 'first peak' (but the cumulative output after each peak is saved?). Should (ii) not be the peak of each 'sub-peak' in a multi-peak storm and should the output not be the volume (for example) between the 2 peaks, rather than the cumulative over the full event? As an aside - Your wave height cutoffs (98 and 99.5%) are also quite high, so you could lower these and get more smaller storms (say the 95% level – see Masselink et al).

**[R2.2]** *With respect to creating a dataset based on storm peaks instead of storms.*
Individual storm events have been identified and isolated by using the P.O.T method that ensures they are independent. Then, from there, any storm consisting in more than one peak is treated by its individual (cumulative) peaks, as the idea was to create a dataset of storm peaks (not to artificially augment the dataset with additional storms). For each peak, we retain its duration, together with the total accumulated event duration, and the previous energy (i.e. single-peak storms are always characterised as peaks with "peak duration" equal to "event duration" and with "zero previous energy"). This was done for a parallel analysis on morphodynamic response where we found that peak sequencing was a key aspect to predict local beach retreats. These variables were included in the network to assess their impact into output risk variables, but for the sake of simplicity only a selection of them, focusing on other variables, is presented here, and due to this they have been shortly described, which could generate some confusion. We have extended the variable description in the revised version.

The reviewer is fully right affirming that each "sub-peak" should be considered (not only the first). In fact, the original dataset contains ALL sub-peaks. Text in L155-160 refers to the fact that

in order to create the subsets for the future scenarios, and with the objective of reducing the number of time-consuming simulations, the first peak of a multipeak storm is also used as a proxy of "single-peak-storms of the same characteristics". We have rephraseed part of the "Storm characterisation" section to clarify this point.

*With respect to threshold selection.*
The used thresholds when applying the P.O.T method (98% and 99.5% percentiles of the wave height distribution) have been previously used in other works in the study area (Sanuy et al., 2019; Sanuy and Jiménez 2020). Obtained results (identified storms) have been compared with storm conditions associated with representative storm classes (Mendoza et al., 2011) and they fit with values obtained therein for Class 1 and Class 3 storms. Class 1 storms have the minimum Hs historically used in the Mediterranean as threshold for extreme events (2 m), while Class 3 events have the minimum Hs that actually induces hazardous coastal response. This is equivalent to define storms as starting and ending with a Class 1 magnitude, and having at least Class 3 at the peak. This permits to assure that all included events will induce a relevant coastal response from the risk-oriented standpoint.
The obtained event density of 3.5 events/year is appropriate for extreme-climate analysis, and lowering the threshold would increase this frequency by including not too extreme events which would not significantly contribute to overall risk. Due to this, we will maintain the proposed thresholds which have been locally validated for this use. In spite of this, we have stressed the meaning of the thresholds, specifying that the levels are site-dependent both in the Storm Characterization" and "Discussion" sections.

**Added text in the manuscript:**
**Storm characterization (L 136):** *The first threshold, the 0.98 quantile (Hs = 2 m, in agreement with Class 1 storms in Mendoza et al. 2011 for NW Mediterranean conditions), is used to identify storm start and end times, and thus, controls the event duration and inter-event fair-weather periods. The second threshold, the 0.995 quantile (Hs = 2.6 m), is used to filter events that do not reach this value at the peak and would not be significant in terms of induced impacts. This second threshold retains only storms reaching Class 3 at the peak which is the minimum storm magnitude inducing hazardous coastal response (Mendoza et al., 2011)*

**Storm characterization (L147):** *For each peak, we retain its duration, together with the total accumulated event duration, and the previous energy (e.g. single-peak storms are always characterised as peaks with "peak duration" equal to "event duration" and with "zero previous energy"). Although all this information is retained (Figure 2), only event duration together with wave parameters and water level will be used as BN variable here, for the sake of simplicity in a risk-oriented perspective, while more detailed source description may be necessary in morphological analyses.*

**Storm characterisation (L159):** *This subdivision is only used for the purpose of deriving the subset, allowing finer detail in the source characteristics of the single-peak and multi-peak*

*storms to be selected. Later, the BN will present a coarser binning of such variables, ensuring a better filling of the source variable combinations in the network.*

**Storm characterisation (L173):** *Thus, to properly account for their potential effects, all existing identified multi-peak storms in the original time-series (43) were included in the subset. Their impact was simulated with the XBeach model saving the cumulative output after each peak. The impact after the first peak of such multi-peak events was used as proxy of equivalent single-peaks already covering 22 source variable combinations. The other 26 combinations where covered by additional single-peak storms.*

**Discussion (L453):** *The thresholds used to identify independent events in the P.O.T are site dependent. In this work, they agree with the storm classification in Mendoza et al., (2011), and therefore they are valid for the Catalan coast (NW Mediterranean)*

Second, my understanding is that inputs to the BNs are meant to be independent, so closely spaced receptors which are highly correlated shouldn't be included. I couldn't find details on the spacing of the receptors, but they don't look spatially independent to me (Eg. Fig 3). Beuzen et al. (2019 – JGR) I think discussed this and found the alongshore spacing allowed where correlations dropped off (This would be site specific but in his case it was _500m I think). So I suspect you've padded your BN with a bunch of data that's highly correlated which isn't best practice.

**[R2.3]** This answer is related with **[R2.5]** (see below). Beuzen et al. (2019-JGR) deals with morphological patterns at regional scale (~400 km). They aim for a predictive BN and therefore they cannot allow for correlations in the input. Indeed, distances would be case specific, and in places as the Tordera Delta (curvilinear shoreline with significant alongshore morphological variability, and beach-structure interactions inducing local processes such as flanking effects) these distances would be much lower, as we found by analysing the morphological response sector by sector (analysed in a companion morphodynamic oriented-paper, currently under review).

However, this is out of the scope of the current paper, which is risk-oriented. Here, the individual receptors must be represented as they indicate the spatial extension and magnitude of the impacts induced by a given coastal response (e.g. its not the same from the risk perspective 100 m of eroded dune in front of 1 receptor than the same 100 m of eroded dune in front of 2 lines of 20 receptors). Thus, we have adopted the Source-Pathway-Receptor-Consequence (SPRC) scheme as in Poelhekke et al (2016), Jäger et al. (2018), Plomaritis et al. (2018) and Sanuy et al. (2018), to account for the actual receptor density and typology at the local scale.

What's the difference between distance to public domain (Fig 7) and distance to beach (Fig 10-12)? I feel they must be similar if not the same so why not use the same classification and binning for the 2?

**[R2.6]** Indeed, they are the same since the line of public domain is running along the inner limit of the beach. The name in figure 7 (and related text) will be changed. The binning is actually the same, but in figures 10-12 the outputs of two lowest bins are summed under the name "Beach to 10m", and the outputs of the four highest bins are summed under the name "> 75 m"

Specific Comments:
[L74]: 'were' should be 'where' in: "study area were"
**[R2.7]** This has been addressed in the updated version of the manuscript.

[L204]: "Risk to life was also been" should be either 'Risk to live was also' or "Risk to life has also been"
**[R2.8]** This has been addressed in the updated version of the manuscript.

Fig 5 - can you tell the reader what section these are in and the erosion rate used?
**[R2.9]** This bas been included in the figure caption

[L355] "affecter" should be 'affected'?
**[R2.10]** This has been addressed in the updated version of the manuscript.

[L341] "front a of a" should be "front of a"
**[R2.11]** This has been addressed in the updated version of the manuscript.

[L427] "relation" should be "relationship
**[R2.12]** This has been addressed in the updated version of the manuscript.

[revised manuscript text omitted]